# PRIVATE STOCHASTIC CONVEX OPTIMIZATION WITH TYSBAKOV NOISE CONDITION AND LARGE LIPSCHITZ CONSTANT

## ABSTRACT

We study Stochastic Convex Optimization in Differential Privacy model (DP-SCO). Unlike previous studies, here we assume the population risk function satisfies the Tysbakov Noise Condition (TNC) with some parameter $\theta > 1$, where the Lipschitz constant of the loss could be extremely large or even unbounded, but the $\ell_2$-norm gradient of the loss has bounded $k$-th moment with $k \geq 2$. For the Lipschitz case with $\theta \geq 2$, we first propose an $(\epsilon, \delta)$-DP algorithms whose utility bound is $\tilde{O}\left( \left( \tilde{r}_{2k}(\frac{1}{\sqrt{n}} + (\frac{\sqrt{d}}{n\epsilon})^{\frac{k-1}{k}}) \right)^{\frac{\theta}{\theta-1}} \right)$ in high probability, where $n$ is the sample size, $d$ is the model dimension, and $\tilde{r}_{2k}$ is a term that only depends on the $2k$-th moment of the gradient. It is notable that such an upper bound is independent of the Lipschitz constant. We then extend to the case where $\theta \geq \bar{\theta} > 1$ for some known constant $\bar{\theta}$. Moreover, when the privacy budget $\epsilon$ is small enough, we show an upper bound of $\tilde{O}\left( \left( \tilde{r}_{k}(\frac{1}{\sqrt{n}} + (\frac{\sqrt{d}}{n\epsilon})^{\frac{k-1}{k}}) \right)^{\frac{\theta}{\theta-1}} \right)$ even if the loss function is not Lipschitz. For the lower bound, we show that for any $\theta \geq 2$, the private minimax rate for $\rho$-zero Concentrated Differential Privacy is lower bounded by $\Omega\left( \left( \tilde{r}_{k}(\frac{1}{\sqrt{n}} + (\frac{\sqrt{d}}{n\sqrt{\rho}})^{\frac{k-1}{k}}) \right)^{\frac{\theta}{\theta-1}} \right)$.

## 1 INTRODUCTION

Machine learning is increasingly being integrated into daily life, driven by an ever-growing volume of data. This data often includes sensitive information, which raises significant privacy concerns. In response, regulations such as the GDPR mandate that machine learning algorithms not only effectively extract insights from training data but also uphold stringent privacy standards. Differential privacy (DP) (12), a robust framework for ensuring statistical data privacy, has garnered substantial attention recently and has emerged as the leading methodology for conducting privacy-preserving data analysis.

Differential Privacy Stochastic Convex Optimization (DP-SCO) and its empirical form, DP Empirical Risk Minimization (DP-ERM), stand as core challenges within the machine learning and differential privacy communities. These methodologies have been the focus of significant research over the past decade, beginning with seminal works like those by Chaudhuri et al. (11) and followed by numerous influential studies (6; 43; 38; 45; 24; 25; 32; 39; 40; 2). For instance, Bassily et al. (5) have provided near-optimal rates for DP-SCO across both convex and strongly convex loss functions. Feldman et al. (14) have developed algorithms that boast linear time complexity, and Su et al. (34) have expanded the discussion to non-Euclidean spaces.

However, the majority of existing theoretical frameworks primarily focus on scenarios where the loss function is $O(1)$-Lipschitz across all data, necessitating assumptions that the underlying data distribution is either bounded or sub-Gaussian. Such assumptions are crucial for the effectiveness of differential privacy methods based on output perturbation (11) and objective or gradient perturbation (6). Yet, these assumptions may not be valid for real-world datasets, particularly those from fields like biomedicine and finance, which are known to exhibit heavy-tailed distributions (44; 7; 19). This discrepancy can compromise the effectiveness of the algorithms in maintaining differential

privacy. To bridge this gap, recent research has begun exploring DP-SCO in the context of heavy-tailed data, where the Lipschitz constant for the loss may be significantly higher or even unbounded (41; 21; 18; 29; 35). These studies typically assume that the gradient of the loss is bounded only in terms of its $k$-th moment for some $k > 0$, a much less stringent requirement than $O(1)$-Lipschitz continuity.

Although DP-SCO with heavy-tailed data has been extensively studied, most research has concentrated on general convex or strongly convex functions. Yet, numerous other problems exist that exceed the complexity of strongly convex functions or do not neatly fit within the convex-to-strongly convex spectrum. In non-private settings, several studies have managed to achieve faster convergence rates by introducing additional constraints on the loss functions. It has been demonstrated that it is possible to exceed the convergence rates of general convex functions (49; 26; 37), and some approaches have even matched the rates typical of strongly convex functions without the function actually being strongly convex (23; 27; 48). Similar advancements have been observed in the context of privacy-preserving algorithms (3; 33). This leads to a compelling question:

**For the problem of DP-SCO with heavy-tailed data and special classes of population risk functions, is it possible to achieve faster rates of excess population risk than the optimal ones of general convex and (or) strongly convex cases?**

In this paper, we affirmatively respond by examining certain classes of population risk functions. Specifically, we focus on the case where the population risk function possesses a large or potentially infinite Lipschitz constant and meets the Tsybakov Noise Condition (TNC) [1], encompassing strongly convex functions, SVM, $\ell_1$-regularized stochastic optimization, and linear regression with heavy-tailed data as notable examples.

Our contributions are detailed as follows (refer to Table 1 for details).

1. We study DP-SCO where the population risk satisfies $(\theta, \lambda)$-TNC with $\theta > 1$. Here, the loss function is $L_f$-Lipschitz, and the $k$-th moment of the loss gradient is small, where $L_f < \infty$ could be extremely large and $k \geq 2$. Based on our newly developed localization method, we propose an $(\epsilon, \delta)$-DP algorithm whose utility bound, with high probability, is $\tilde{O}((\tilde{r}_{2k}(\frac{1}{\sqrt{n}} + (\frac{\sqrt{d}}{n\epsilon}))^{\frac{k-1}{k}})^{\frac{\theta}{\theta-1}})$ when $\theta \geq 2$. Here, $n$ is the sample size, $d$ is the model dimension and $\tilde{r}_{2k}$ is a term that only depends on the $2k$-th moment of the gradient. It is notable that such an upper bound is independent of the Lipschitz constant.

2. We further relax the assumption that $\theta \geq 2$ to $\theta \geq \bar{\theta} > 1$ for some known $\bar{\theta}$ and propose an algorithm that could achieve asymptotically the same bound as the previous one. Moreover, when the privacy budget $\epsilon$ is small enough, we show that even if the loss function is not Lipschitz, we can still get an upper bound of $\tilde{O}((\tilde{r}_k(\frac{1}{\sqrt{n}} + (\frac{\sqrt{d}}{n\epsilon}))^{\frac{k-1}{k}})^{\frac{\theta}{\theta-1}})$.

3. On the lower bound side, for any $\theta \geq 2$, we show that there exists a population risk function satisfying TNC with parameter $\theta$, whose minimax population risk under $\rho$-zero Concentrated Differential Privacy is always lower bounded by $\Omega((\tilde{r}_k(\frac{1}{\sqrt{n}} + (\frac{\sqrt{d}}{n\sqrt{\rho}}))^{\frac{k-1}{k}})^{\frac{\theta}{\theta-1}})$.

## 2 RELATED WORK

**DP-SCO with Heavy-tailed Data.** As we mentioned previously, there is a long list of work for DP-SCO from various perspectives. Here we only focus on the work related to DP-SCO with heavy-tailed data. Generally speaking, there are two ways of modeling heavy-tailedness: The first one considers each coordinate of loss gradient has bounded moments, while the second one assumes the norm of loss gradient has bounded moments, which is stronger than the first one. For the first direction, (41) provides the first study under the assumption of bounded $k$-th moment ($k \geq 2$) and proposes three different ways for both convex and strongly convex loss. The bounds were later improved by (20). Specifically, (20) provides improved upper bounds for convex loss and optimal rate for strongly convex loss. Later, there are some works that consider different extensions. For example, (18) extends to the high dimensional and polyhedral cases, (35) extends to the case where the gradient only has $(1 + v)$-th moment with $v \in (0, 1]$, (42) considers the $\ell_1$-regression. For the second direction, (29)

---

[1]This is also referred to as the Error Bound Condition or the Growth Condition in related literature (27; 48).

Table 1: Comparion with previous results on DP-SCO with different assumptions in $(\epsilon, \delta)$-DP (we always assume the loss is smooth). All results omit the term of $\log\frac{1}{\delta}$, smoothness and strong convexity. † means the result is for $\rho$-zCDP. ⋆ indicated the result holds when $\epsilon = \tilde{O}(\frac{1}{n})$.

| | Upper Bound | Lower Bound | Assumption |
|---|---|---|---|
| (5) | $O\left(\frac{1}{\sqrt{n}} + \frac{\sqrt{d}}{n\epsilon}\right)$ | $\Omega\left(\frac{1}{\sqrt{n}} + \frac{\sqrt{d}}{n\epsilon}\right)$ | $O(1)$-Lipschitz |
| (5) | $O\left(\frac{1}{n} + \frac{d}{n^2\epsilon^2}\right)$ | $\Omega\left(\frac{1}{n} + \frac{d}{n^2\epsilon^2}\right)$ | $O(1)$-Lipschitz |
| (20) | $\tilde{O}\left(\frac{d}{\sqrt{n}} + \frac{d^2}{n\epsilon}(\frac{\epsilon n}{d^{\frac{3}{2}}})^{\frac{1}{k}}\right)$ | $\Omega(\sqrt{\frac{d}{n}} + \sqrt{d}(\frac{\sqrt{d}}{n\sqrt{\rho}})^{\frac{k-1}{k}})^{\dagger}$ | $O(1)$-Lipschitz and bounded $k$-th moment ($k \geq 2$) |
| (20) | $\tilde{O}\left(\frac{d}{n} + d(\frac{\sqrt{d}}{n\epsilon})^{\frac{2(k-1)}{k}}\right)$ | $\Omega\left(\frac{d}{n} + d(\frac{\sqrt{d}}{n\sqrt{\rho}})^{\frac{2(k-1)}{k}}\right)^{\dagger}$ | $O(1)$-Lipschitz, strongly convex and bounded $k$-th moment ($k \geq 2$) |
| (2; 33) | $\tilde{O}\left(\left(\frac{1}{\sqrt{n}} + \frac{\sqrt{d}}{n\epsilon}\right)^{\frac{\theta}{\theta-1}}\right)$ | $\Omega\left(\left(\frac{1}{\sqrt{n}} + \frac{\sqrt{d}}{n\epsilon}\right)^{\frac{\theta}{\theta-1}}\right)$ when $\theta \geq 2$ | $O(1)$-Lipschitz under TNC with $\theta > 1$ |
| (29) | $O\left(\widetilde{R}_{2k,n}\left(\frac{1}{\sqrt{n}} + \left(\frac{\sqrt{d}}{\epsilon n}\right)^{\frac{k-1}{k}}\right)\right)$ | $\Omega\left(\tilde{r}_k\left(\frac{1}{\sqrt{n}} + \left(\frac{\sqrt{d}}{\sqrt{\rho}n}\right)^{\frac{k-1}{k}}\right)\right)^{\dagger}$ | (large) Lipschitz, bounded $k$-th moment ($k \geq 2$) |
| (29) | $\tilde{O}\left(\tilde{r}_k\left[\frac{1}{\sqrt{n}} + \max\left\{\left(\left(\frac{1}{\tilde{r}_k}\right)^{1/4}\frac{\sqrt{d}}{\epsilon n}\right)^{\frac{4(k-1)}{5k-1}}, \left(\frac{\sqrt{d}}{\epsilon n}\right)^{\frac{k-1}{k}}\right\}\right]\right)$ | $\Omega\left(\tilde{r}_k\left(\frac{1}{\sqrt{n}} + \left(\frac{\sqrt{d}}{\sqrt{\rho}n}\right)^{\frac{k-1}{k}}\right)\right)^{\dagger}$ | bounded $k$-th moment ($k \geq 2$) |
| (29) | $\tilde{O}\left(\widetilde{R}_{2k,n}^2\left(\frac{1}{n} + \left(\frac{\sqrt{d}}{\epsilon n}\right)^{\frac{2(k-1)}{k}}\right)\right)$ | $\Omega\left(\tilde{r}_k^2\left(\frac{1}{n} + \left(\frac{\sqrt{d}}{\sqrt{\rho}n}\right)^{\frac{2(k-1)}{k}}\right)\right)^{\dagger}$ | strongly convex, bounded $k$-th moment ($k \geq 2$) |
| This paper | $\tilde{O}\left(\left(\widetilde{R}_{2k,n}(\frac{1}{\sqrt{n}} + \left(\frac{\sqrt{d}}{\epsilon n}\right)^{\frac{k-1}{k}})\right)^{\frac{\theta}{\theta-1}}\right)$ | $\Omega\left(\left(\tilde{r}_k(\frac{1}{\sqrt{n}} + \left(\frac{\sqrt{d}}{\epsilon n}\right)^{\frac{k-1}{k}})\right)^{\frac{\theta}{\theta-1}}\right)$ when $\theta \geq 2$ | (large) Lipschitz function under TNC with $\theta > 1$ |
| This paper | $\tilde{O}\left(\tilde{r}_k^{\frac{\theta}{\theta-1}}\left(\frac{1}{\sqrt{n}} + \left(\frac{\sqrt{d}}{\epsilon n}\right)^{\frac{k-1}{k}}\right)^{\frac{\theta}{\theta-1}}\right)^{\star}$ | $\Omega\left(\left(\tilde{r}_k(\frac{1}{\sqrt{n}} + \left(\frac{\sqrt{d}}{\epsilon n}\right)^{\frac{k-1}{k}})\right)^{\frac{\theta}{\theta-1}}\right)$ when $\theta \geq 2$ | TNC with $\theta > 1$ |

provides a comprehensive study for both convex and strongly convex loss. In detail, for Lipschitz loss whose gradient has $k$-th moment, they provide upper bounds that are independent of the Lipschitz constant. Compared to (29), we first extend to the population risks that satisfy TNC (when $\theta = 2$, our results match their results for strongly convex loss). Moreover, the results in (29) are in expectation form while we provide new algorithms, and our results are in the high probability form.

**DP for Heavy-tailed Data.** In addition to DP-SCO, there is also some work on DP for heavy-tailed data. (4) provided the first study on private mean estimation for distributions with the bounded moment, which has been extended by (22; 28; 9) recently. However, these methods cannot be applied to our problem as these results are all in the expectation form. Motivated by (41), we later consider statistical guarantees of DP Expectation Maximization and applies to the Gaussian Mixture Model. (46; 36; 47) considers private reinforcement learning and bandits learning where the reward follows a heavy-tailed distribution. However, since the reward is a scalar, these methods are not applicable to our problem.

**Loss functions with TNC.** While most of this paper focuses on loss functions that are either convex or strongly convex, many loss functions fall between these two categories. That is, they are not strongly convex, but their statistical rate is better than purely convex losses. For TNC and Lipschitz loss functions, the best-known current rate is $O((\frac{1}{\sqrt{n}})^{\frac{\theta}{\theta-1}})$ (27), which corresponds to the first term in our upper bounds. In Theorem 5, we demonstrate that this upper bound is tight for $\theta \geq 2$. A comparison to the non-private setting will be included in the final version of the paper.

Our methods introduce novel technical challenges compared to non-private approaches. The key innovation lies in our analysis, which is based on algorithmic stability and a newly developed localized and clipped algorithm (Algorithm 3), which has not been previously studied. Specifically, Algorithm 4 is inspired by Algorithm 2 in (27). However, while the base algorithm in (27) is a simple averaged version of projected SGD, our Algorithm 3 is significantly more complex. One major technical challenge is that Algorithm 2 in (27) assumes a Lipschitz loss function with a fixed Lipschitz constant. Consequently, their bounds rely on this constant. In contrast, we address scenarios where the Lipschitz parameter can be extremely large. Therefore, we developed a new base algorithm that removes dependence on this parameter and instead utilizes moments.

## 3 PRELIMINARIES

**Definition 1** (Differential Privacy (12)). Given a data universe $\mathcal{X}$, we say that two datasets $S, S' \subseteq \mathcal{X}$ are neighbors if they differ by only one entry, which is denoted as $S \sim S'$. A randomized algorithm $\mathcal{A}$ is $(\epsilon, \delta)$-differentially private (DP) if for all neighboring datasets $S, S'$ and for all events $E$ in the

output space of $\mathcal{A}$, the following holds

$$\mathbb{P}(\mathcal{A}(S) \in E) \leqslant e^{\epsilon}\mathbb{P}(\mathcal{A}(S') \in E) + \delta.$$

If $\delta = 0$, we call algorithm $\mathcal{A}$ is $\epsilon$-DP.

**Definition 2** (zCDP (10)). A randomized algorithm $\mathcal{A}$ is $\rho$-zero-concentrate-differentially private (zCDP) if for all neighboring datasets $S, S'$ and $\alpha \in (1, \infty)$, we have $D_{\alpha}(\mathcal{A}(S)\|\mathcal{A}(S')) \leqslant \rho\alpha$, where $D_{\alpha}$ is the $\alpha$-Rényi divergence between $\mathcal{A}(S)$ and $\mathcal{A}(S')$.

**Remark 1.** In this paper, we focus on $(\epsilon, \delta)$-DP for upper bounds and $\rho$-zCDP for lower bounds, and we mainly use the Gaussian mechanism to guarantee the DP property. For Algorithms 1-5, which are based on stability analysis and the Gaussian mechanism, they operate as one-pass algorithms without sub-sampling. As a result, they can easily meet the requirements for CDP. However, a challenge arises with Algorithm 6. In this case, we employ privacy amplification via shuffling to reduce the noise. Currently, privacy amplification via shuffling is only applicable to $\epsilon$ and $(\epsilon, \delta)$-LDP, and no version exists for zCDP. To maintain consistency throughout the paper, we use $(\epsilon, \delta)$-DP for all our upper bounds.

**Definition 3** (Gaussian Mechanism). Given any function $q : \mathcal{X}^n \to \mathbb{R}^d$, the Gaussian mechanism is defined as $q(S) + \xi$ where $\xi \sim \mathcal{N}(0, \frac{16\Delta_2^2(q)\log(1/\delta)}{\epsilon^2}\mathbb{I}_d)$, where $\Delta_2(q)$ is the $\ell_2$-sensitivity of the function $q$, *i.e.*, $\Delta_2(q) = \sup_{S \sim S'} \|q(S) - q(S')\|_2$. Gaussian mechanism preserves $(\epsilon, \delta)$-DP for $0 < \epsilon, \delta \leqslant 1$.

**Definition 4** (DP-SCO (6)). Given a dataset $S = \{x_1, \cdots, x_n\}$ from a data universe $\mathcal{X}$ where $x_i$ are i.i.d. samples from some unknown distribution $\mathcal{D}$, a convex loss function $f(\cdot, \cdot)$, and a convex constraint set $\mathcal{W} \subseteq \mathbb{R}^d$, Differentially Private Stochastic Convex Optimization (DP-SCO) is to find $w^{\text{priv}}$ so as to minimize the population risk, *i.e.*, $F(w) = \mathbb{E}_{x \sim \mathcal{D}}[f(w, x)]$ with the guarantee of being differentially private. The utility of the algorithm is measured by the *(expected) excess population risk*, that is

$$\mathbb{E}_{\mathcal{A}}[F(w^{\text{priv}})] - \min_{w \in \mathcal{W}} F(w),$$

where the expectation of $\mathcal{A}$ is taken over all the randomness of the algorithm. Besides the population risk, we may also measure the *empirical risk* of dataset $S$: $\bar{F}(w, S) = \frac{1}{n}\sum_{i=1}^{n} f(w, x_i)$.

**Definition 5** (Lipschitz). A function $f : \mathcal{W} \mapsto \mathbb{R}$ is L-Lipschitz over the domain $\mathcal{W}$ if for all $w, w' \in \mathcal{W}$, $|f(w) - f(w')| \leqslant L\|w - w'\|_2$.

**Definition 6** (Smoothness). A function $f : \mathcal{W} \mapsto \mathbb{R}$ is $\beta$-smooth over the domain $\mathcal{W}$ if for all $w, w' \in \mathcal{W}$, $f(w) \leqslant f(w') + \langle \nabla f(w'), w - w' \rangle + \frac{\beta}{2}\|w - w'\|_2^2$.

**Definition 7** (Strongly Convex). A function $F : \mathcal{W} \mapsto \mathbb{R}$ is $\lambda$-strongly convex over the domain $\mathcal{W}$ if, for all $w, w' \in \mathcal{W}$, $F(w) + \langle \nabla F(w), w' - w \rangle + \frac{\lambda}{2}\|w' - w\|_2^2 \leqslant F(w')$.

Previous work on DP-SCO only focused on cases where the loss function is either convex or strongly convex (5; 14). In this paper, we mainly examine the case where the population risk satisfies the Tysbakov Noise Condition (TNC) (30; 27; 31), which has been extensively studied and has been shown that it could achieve faster rates than the optimal one of general convex loss functions in the non-private case. Below, we introduce the definition of TNC.

**Definition 8** (Tysbakov Noise Condition). For a convex function $F(\cdot)$, let $\mathcal{W}_* = \arg\min_{w \in \mathcal{W}} F(w)$ denote the optimal set and for any $w \in \mathcal{W}$, let $w^* = \arg\min_{u \in \mathcal{W}_*} \|u - w\|_2$ denote the projection of $w$ onto the optimal set $\mathcal{W}_*$. The function $F$ satisfies $(\theta, \lambda)$-TNC for some $\theta > 1$ and $\lambda > 0$ if, for any $w \in \mathcal{W}$, the following inequality holds:

$$F(w) - F(w^*) \geq \lambda\|w - w^*\|_2^{\theta}. \tag{1}$$

From the definition of TNC and Definition 7, we can see that a $\lambda$-strong convex function is $(2, \frac{\lambda}{2})$-TNC. Moreover, if a function is $(\theta, \lambda)$-TNC, then it is also $(\theta', \lambda)$-TNC for any $\theta < \theta'$. Throughout the paper, we assume that $\theta$ is a constant and thus we omit the term of $c^{\theta}$ in the Big-$O$ notation if $c$ is a constant.

**Lemma 1** (Lemma 2 in (30)). *If the function $F(\cdot)$ is $(\theta, \lambda)$-TNC and $L_f$-Lipschitz, then we have $\|w - w^*\|_2 \leqslant (L_f\lambda^{-1})^{\frac{1}{\theta-1}}$ and $F(w) - F(w^*) \leqslant (L_f^{\theta}\lambda^{-1})^{\frac{1}{\theta-1}}$ for all $w \in \mathcal{W}$, where $w^*$ is defined as in Definition 8.*

As mentioned earlier, our primary focus here is on cases where the loss function's Lipschitz constant is sufficiently large or even infinite. In such cases, we may seek alternative terms to replace the Lipschitz constant. Motivated by previous work on DP-SCO with heavy-tailed gradients, we consider the moments of the gradient. Specifically, we assume that the stochastic gradient distributions have bounded $k$-th moment for some $k \geqslant 2$:

**Assumption 1.** There exists $k \geqslant 2$ and $\widetilde{r}^{(k)} > 0$ such that $\mathbb{E}\left[\sup_{w \in \mathcal{W}} \|\nabla f(w, x)\|_2^k\right] \leqslant \widetilde{r}^{(k)}$, where $\widetilde{r}_k := \left(\widetilde{r}^{(k)}\right)^{1/k}$. Moreover, we assume the constrained set $\mathcal{W}$ is bounded with diameter $D$.

If the loss function is $L_f$-Lipschitz, we can always observe that $\widetilde{r}_k \leqslant L_f = \sup_{w,x} \|\nabla f(w, x)\|_2$. Moreover, $\widetilde{r}_k$ could be far less than the Lipschitz constant.

To state our subsequent theoretical results more clearly, we introduce some additional notations. For a batch of data $X \in \mathcal{X}^m$, we define the $k$-th empirical moment of $f(w, \cdot)$, by

$$\widehat{r}_m(X)^{(k)} = \sup_{w \in \mathcal{W}} \frac{1}{m} \sum_{i=1}^{m} \|\nabla f(w, x_i)\|_2^k.$$

For $X \sim \mathcal{D}^m$, we denote the $k$-th expected empirical moment by

$$\widetilde{e}_m^{(k)} := \mathbb{E}[\widehat{r}_m(X)^{(k)}],$$

and let

$$\widetilde{r}_{k,m} := (\widetilde{e}_m^{(k)})^{1/k}.$$

Note that $\widetilde{r}_{k,1} = \widetilde{r}_k$. We define $\widetilde{R}_{k,n} := \sqrt{\sum_{i=1}^{l} 2^{-i} \widetilde{r}_{k,n_i}^2}$, where $n_i = 2^{-i} n$ and $l = \log_2 n$. Actually, $\widetilde{R}_{k,n}$, a weighted average of the expected empirical moments for distinct batch sizes, is a constant used in the excess risk upper bounds, where we give more weight to $\widetilde{r}_m$ for large $m$. The following lemma indicates that it is smaller than $\widetilde{r}_k$.

**Lemma 2** ((29)). *Under Assumption 1, we have: $\widetilde{r}^{(k)} = \widetilde{e}_1^k \geqslant \widetilde{e}_2^k \geqslant \widetilde{e}_4^k \geqslant \cdots \geqslant r^{(k)}$. Thus, we have* $\widetilde{R}_{k,n} \leqslant \widetilde{r}_k$.

## 4 LARGE LIPSCHITZ CONSTANT CASE

In this section, we will focus on the population risk function satisfying $(\theta, \lambda)$-TNC, and the Lipschitz constant of the loss is extremely large (but finite). Before that, we first propose a novel localized noisy stochastic gradient method whose excess population risk is independent of the Lipschitz constant for general convex loss. See Algorithm 3 for details.

In Algorithm 3, we first partition the dataset into $O(\log_2 n)$ subsets where the $i$-th set has $O(2^{-i}n)$ samples. In the $i$-th iteration, we use the $i$-th set and construct an $\ell_2$-regularized empirical risk function $F_i$ with hyperparameter $\lambda_i$ in step 5. Moreover, based on the current model $w_{i-1}$, we construct the constrained set $\mathcal{W}_i$ with diameter exponential decay $D_i$. To handle large Lipschitz constant and to solve the $\ell_2$-regularized empirical risk, we adopt a clipped gradient descent method (Algorithm 2) with clip threshold $C_i$, where we use clipped gradients (Algorithm 1) to update our model instead of the original gradient. After $T_i$ iterations, we add Gaussian noise based on the stability of our clipped gradient descent to ensure $(\epsilon, \delta)$-DP. In the following we show Algorithm 3 could achieve a rate $\widetilde{O}(\max\{\frac{1}{\sqrt{n}}, (\frac{d \log \frac{1}{\delta}}{\epsilon n})^{\frac{k-1}{k}}\})$ with specific parameters $\lambda_i, T_i$ and $C_i$.

---

**Algorithm 1** ClippedMean($\{z_i\}_{i=1}^n, n, C$)

**Input:** $Z = \{z_i\}_{i=1}^n, C > 0$,

1: Compute $\widetilde{v} := \frac{1}{n} \sum_{i=1}^{n} \prod_C(z_i)$, where $\prod_C(z) := \operatorname{argmin}_{y \in \mathbb{B}_C} \|y - z\|_2^2$ denotes the projection onto the $\ell_2$ ball $\mathbb{B}_C$.

**Return** $\widetilde{v}$

---

---

**Algorithm 2** Clipped Regularized Gradient Method

---

**Input:** Dataset $S \in \mathcal{X}^n$, iteration number $T$, stepsize $\eta$, clipping threshold $C$, regularization $\lambda \geqslant 0$, constraint set $\mathcal{W}$ and initialization $w_0 \in \mathcal{W}$.

1: **for all** $t \in [T-1]$ **do**
2:     $\widetilde{\nabla} F_t(w_t) :=$ ClippedMean($\{\nabla f(w_t, x_i)\}_{i=1}^n; C$) for gradients $\nabla f(w_t, x_i)$.
3:     $w_{t+1} = \prod_{\mathcal{W}}[w_t - \eta(\widetilde{\nabla} F_t(w_t) + \lambda(w_t - w_0))]$
4: **end for**
**Return** $w_T$

---

**Theorem 1.** *Under Assumption 1, suppose that $f(\cdot, x)$ is $\alpha$-smooth and $L_f$-Lipschitz with $L_f < \infty$ for every $x$. Then, for any $0 < \epsilon \leqslant \sqrt{\log(1/\delta)}, 0 < \delta < 1$ and $\eta_i \leqslant \frac{1}{\alpha}$ for all $i$, Algorithm 3 is $(\epsilon, \delta)$-DP. Let $p \geq 1$ such that $L_f \leqslant n^{p/2} \widetilde{R}_{2k,n}(\frac{1}{\sqrt{n}} + (\frac{\sqrt{d \log n}}{\epsilon n})^{\frac{k-1}{k}})$. For any $0 < \beta \leqslant \frac{1}{n}$, with probability at least $1 - \beta$, it holds that*

$$F(w_l) - F(w^*) \leqslant \tilde{O}\left(\widetilde{R}_{2k,n} D(\frac{1}{\sqrt{n}} + (\frac{\sqrt{d}}{\epsilon n})^{\frac{k-1}{k}})\right),$$

*where the Big-$\tilde{O}$ notation omits all logarithmic terms (it is the same for other upper bounds).*

**Remark 2.** Previous work on DP-SCO such as (43; 6), Lipschitz is still required for the loss function, though, it disappears in the final excess risk upper bound. And due to the property of worst-case stability and our assumption that $L_f$ can be controlled by $n^{p/2} \widetilde{R}_{2k,n}(\frac{1}{\sqrt{n}} + (\frac{\sqrt{d \log n}}{\epsilon n})^{\frac{k-1}{k}})$ for sufficiently large $p$, we reach the upper bound with high probability without $L_f$ in the final result. Compared to (29), the main difference is that our result is in the high probability form while (29) is only in the expectation form. Specifically, to achieve a high probability result, instead of adding Gaussian noise to the gradient, we use the stability of the gradient descent. However, we cannot directly use the stability result in (17) here, which depends on the Lipschitz constant, making a large noise, we show that by using clipping, the stability now only depends on the clipping threshold.

---

**Algorithm 3** Localized Noisy Clipped Gradient Method for DP-SCO(LNC-GM)$(w_0, \eta, n, \mathcal{W})$

---

**Input:** Dataset $S \in \mathcal{X}^n$, stepsize $\eta$, clipping threshold $\{C_i\}_{i=1}^{\log_2 n}$, privacy parameter $\epsilon, \delta$, hyperparameter $p$, initialization $w_0 \in \mathcal{W}$.

1: Let $l = \log_2 n$.
2: **for all** $i \in [l]$ **do**
3:     Set $n_i = 2^{-i} n, \eta_i = 4^{-i} \eta$, $\lambda_i = \frac{1}{\eta_i n_i^p}$ when $i \geqslant 2$, and $\lambda_1 = \frac{1}{\eta_1 n_1^{2p}}$, $T_i = \widetilde{\Theta}\left(\frac{1}{\lambda_i \eta_i}\right)$, and $D_i = \frac{2L_f}{\lambda_i}$.
4:     Draw a new batch $\mathcal{B}_i$ of $n_i = |\mathcal{B}_i|$ samples from $S$ without replacement.
5:     Denote $\widehat{F}_i(w) := \frac{1}{n_i} \sum_{j \in \mathcal{B}_i} f(w, x_j) + \frac{\lambda_i}{2} \|w - w_{i-1}\|^2$.
6:     Use Algorithm 2 with initialization $w_{i-1}$ to minimize $\widehat{F}_i$ over $\mathcal{W}_i := \{w \in \mathcal{W} | \|w - w_{i-1}\| \leqslant D_i\}$ for $T_i$ iterations with clipping threshold $C_i = \tilde{r}_{2k,n_i}(\frac{\epsilon n_i}{\sqrt{d \log(n)}})^{1/k}$ and stepsize $\eta_i$. Let $\hat{w}_i$ be the output of Algorithm 2.
7:     Set $\xi_i \sim \mathcal{N}(0, \sigma_i^2 \mathbb{I}_d)$ where $\sigma_i = \frac{8C_i \sqrt{\log \frac{1}{\delta}}}{n_i \lambda_i \epsilon}$
8:     Set $w_i = \widehat{w}_i + \xi_i$.
9: **end for**
10: **Return** the final iterate $w_l$

---

Based on our novel locality algorithm, we then apply it to TNC functions. See Algorithm 4 for details. Specifically, we partition the dataset into several subsets of equal size. As the iteration number increases, we consider a constrained set centered at the current parameter with a smaller diameter and learning rate in Algorithm 3.

**Theorem 2.** *Under Assumption 1 and suppose that the population risk function $F(\cdot)$ is $(\theta, \lambda)$-TNC with $\theta \geq 2$, and $f(\cdot, x)$ is $\alpha$-smooth and $L_f$-Lipschitz for each $x$. Additionally, take $p \geq 1$ such that*

---

**Algorithm 4** Private Stochastic Approximation$(w_1, n, R_0)$

---

**Input:** Dataset $S \in \mathcal{X}$, initial point $w_1 \in \mathcal{W}$, privacy parameter $\epsilon$ and $\delta$, hyperparameter $p$, initial diameter $R_0$.

1: Set $\hat{w}_0 = w_1$, $m = \lfloor \frac{1}{2} \log_2 \frac{2n}{\log_2 n} \rfloor - 1$, $n_0 = \lfloor \frac{n}{m} \rfloor$. Then partition the dateset $S$ into $m$ disjoint subsets, namely, $\{S_1, \cdots, S_m\}$ with each $|S_i| = n_0$.

2: **for all** $l \in [m]$ **do**

3: Set $\gamma_l = \frac{R_{l-1}}{n_0^{\frac{p}{2}}} \min\{\frac{1}{L_f}, \frac{1}{\widetilde{R}_{2k,n} n_0^{\frac{p+1}{2}}} (\frac{\epsilon n_0}{\sqrt{d \log n}})^{\frac{k-1}{k}}, \frac{1}{n_0^{\frac{p-1}{2}} L_f^2 \sqrt{\log n_0 \log(1/\beta)}}\}$ and $R_l = \frac{R_{l-1}}{2}$.

4: Denote $\hat{w}_l = \text{LNC-GM}(\hat{w}_{l-1}, \gamma_l, n_l, \mathcal{W})$, and constrained set $\mathcal{W} \cap \mathbb{B}(\hat{w}_{l-1}, R_{l-1})$.

5: **end for**

**Return** $\hat{w}_m$

---

$L_f \leqslant n^{p/2} \widetilde{R}_{2k,n}(\frac{1}{\sqrt{n}} + (\frac{\sqrt{d \log n}}{\epsilon n})^{\frac{k-1}{k}})$, *then algorithm 4 is* $(\epsilon, \delta)$-*DP. Moreover, for sufficiently large* $n$ *such that* $\gamma_l \leqslant \frac{1}{\alpha}$, *with probability at least* $1 - \beta$, *we have*

$$F(\hat{w}_m) - F(w^*) \leqslant \tilde{O}\left(\frac{1}{\lambda^{\frac{1}{\theta-1}}} (\widetilde{R}_{2k,n}(\frac{1}{\sqrt{n}} + (\frac{\sqrt{d}}{\epsilon n})^{\frac{k-1}{k}}))^{\frac{\theta}{\theta-1}}\right).$$

We note that there is no dependence on p in the final bound in Theorem 1 and 2. $p$ is used to control the Lipschitz constant thus we can remove the Lipschitz constant from the final bound. We can see that in the proof of Theorem 1, there exists a term with $n^p$ both in the numerators and denominators. By assuming that $L_f$ is controlled by the $O(n^p/2)$ and choosing specific $\eta$, we can eliminate the $p$ in the final bound. A similar result holds for Theorem 2.

**Remark 3.** In the case of $O(1)$-Lipschitz loss under TNC, compared with the optimal rate $\Theta(((\frac{1}{\sqrt{n}} + (\frac{\sqrt{d}}{\epsilon n})^{\frac{k-1}{k}}))^{\frac{\theta}{\theta-1}})$ in (3), our improved result gets rid of the dependence of Lipschitz constant, which could be extremely large. Moreover, when $\theta = 2$, i.e., the population risk is strongly convex, our result covers the result in (29). Thus, our result is a generalized upper bound. It is also notable that our upper bound is independent of the diameter of the constrained set and the Lipschitz-smoothness parameter. In Algorithm 4, one need the projection onto the ball $\mathcal{W} \cap \mathbb{B}(\hat{w}_{l-1}, R_{l-1})$ in each iteration of the Phased-SGD in each phase. This could be solved using Dykstra's algorithm (13; 8).

**Example.** We consider the $\ell_1$ constrained $\ell_4$-norm linear regression, which has been studied in (48) and satisfies TNC with $\theta = 4$ (27). Specifically, it can be written as the following.

$$\min_{\|w\|_1 \leqslant 1} F(w) \triangleq \mathbb{E}[(\langle w, x \rangle - y)^4]. \tag{2}$$

When $y$ is bounded by $O(1)$ and $x$ follows a truncated normal Gaussian distribution at $[-n, n]^d$. Then we can see that the loss function is $\text{Poly}(n)$-Lispchitz, but its $2k$-th moment is $O(1)$. In this case, our bound in equation 2 is much smaller than the previous results in (3; 33).

So far, we have proposed an algorithm for TNC. Nevertheless, we also find that Theorem 2 needs a strong assumption on $\theta$, i.e. $\theta \geqslant 2$. Thus, a direct question that occurs to us is whether we can further improve the upper bound. To conquer the disadvantage of the above algorithm, we propose the following. We assume $\theta$ is unknown but bigger than some definite $\bar{\theta} > 1$. Then we divide the whole dataset into subsets with distinct elements, detailly $l = \lfloor (\log_{\bar{\theta}} 2) \cdot \log \log n \rfloor$ with $n_i = \lfloor 2^{i-1} n/(\log n)^{\log_{\bar{\theta}}^2 2} \rfloor$ samples for each subset. Then we run the Algorithm 1 for $l$ times while each phase implements on the $i$-th subset and is initialized at the output of the previous one.

**Theorem 3.** *Under Assumption 1 and assume that the loss function* $F(\cdot)$ *satisfies* $(\theta, \lambda)$-*TNC with parameter* $\theta \geqslant \bar{\theta} > 1$ *for some definite constant* $\bar{\theta}$, *and* $f(\cdot, x)$ *is convex,* $\alpha$ *smooth and* $L_f$-*Lipschitz for each* $x$. *Algorithm 5 is* $(\epsilon, \delta)$-*DP for any* $\epsilon \leqslant 2 \log(1/\delta)$, *and take* $p \geq 1$ *such that* $L_f \leqslant n^{p/2} \widetilde{R}_{2k,n}(\frac{1}{\sqrt{n}} + (\frac{\sqrt{d \log n}}{\epsilon n})^{\frac{k-1}{k}})$. *Moreover, if the sample size* $n$ *is sufficiently large such that* $\bar{\theta} \geqslant 2^{\frac{\log \log n}{\log n - 1}}$ *and* $\eta_t \leqslant \frac{1}{\alpha}$, *we have with probability at least* $1 - \beta$

$$F(w_l) - F(w^*) \leqslant \tilde{O}\left((\frac{1}{\lambda})^{\frac{1}{\theta-1}} (\widetilde{R}_{2k,n}(\frac{1}{\sqrt{n}} + (\frac{\sqrt{d}}{\epsilon n})^{\frac{k-1}{k}}))^{\frac{\theta}{\theta-1}}\right),$$

---

**Algorithm 5** Iterated Localized Noisy Clipped Gradient Method

---

**Input:** Dataset $S \in \mathcal{X}^n$, initial point $w_0 \in \mathcal{W}$, privacy parameter $\epsilon$ and $\delta$, parameter $p$, initial diameter $R_0$.

1: Partite the data $S$ into $l$ disjoint subsets $\{S_1, \cdots, S_l\}$, where $l = \lfloor (\log_{\bar{\theta}} 2) \cdot \log \log n \rfloor$ and for each $i \in [l], |S_i| = n_i = \left\lfloor 2^{i-1} n / (\log n)^{\log_{\bar{\theta}}^2 2} \right\rfloor$.

2: **for all** $t = 1, \cdots, l$ **do**

3:      Let $w_t$ = Algorithm 3 with input $(S_i, w_{t-1}, \eta_t, \mathcal{W})$, where $\eta_t = \frac{R_{t-1}}{n_0^{\frac{p}{2}}} \min\{\frac{1}{L_f}, \frac{1}{\widetilde{R}_{2k,n} n_i^{\frac{p+1}{2}}} \left(\frac{\epsilon n_i}{\sqrt{d \log n}}\right)^{\frac{k-1}{k}}, \frac{1}{n_i^{\frac{p-1}{2}} L_f^2 \sqrt{\log n_i \log(1/\beta)}}\}$ and $R_l = \frac{R_{l-1}}{2}$.

4: **end for**

**Return** $w_k$

---

**Remark 4.** We pause to have another glimpse of Algorithm 4 and Algorithm 5. Note that they have a similar procedure to take the dataset apart, while the number of each subset is the same in Algorithm 5 and increases in Algorithm 5 as the iteration grows. And the set we project on also varies between Algorithm 4 and 5.

## 5 LOWER BOUNDS

In this section, we will show that the above upper bounds is nearly optimal (if $\tilde{r}_{2k}$ and $\tilde{r}_k$ are asymptotically the same) by providing lower bounds of the private minimax rate for $\rho$-zCDP. Specifically, for a sample space $\mathcal{X} \subseteq \mathbb{R}^d$ and collection of distributions $\mathcal{P}$ over $\mathcal{X}$, we define the function class $\mathcal{F}_k^\theta(\mathcal{P}, \tilde{r}^{(k)})$ as the set of population risk functions from $\mathbb{R}^d \mapsto \mathbb{R}$ that satisfy $(\theta, 1)$-TNC and their loss satsifies Assumption 1. We define the constrained minimax risk

$$\mathcal{M}(\mathcal{W}, \mathcal{P}, \mathcal{F}_k^\theta(\mathcal{P}, \tilde{r}^{(k)}), \rho) = \inf_{\mathcal{A} \in \mathcal{Q}(\rho)} \max_{F \times P \in \mathcal{F}_k^\theta(\mathcal{P}, \tilde{r}_k) \times \mathcal{P}} \mathbb{E}_{\mathcal{A}, D \in P^n}[F(\mathcal{A}(D)) - \min_{w \in \mathcal{W}} F(w)],$$

where $\mathcal{Q}(\rho)$ is the set of all $\rho$-zCDP algorithms. We will show the following two results for different sample spaces and constraint sets.

**Theorem 4.** For any $\theta, k \geq 2, \tilde{r}^{(k)} > 0$, denote $\mathcal{X} = \{\pm p^{-\frac{1}{k}} \frac{\tilde{r}_k}{2\sqrt{d}}\}^d \cup \{0\}$ with $\tilde{r}_k = (\tilde{r}^{(k)})^{\frac{1}{k}}$, and $\mathcal{W} = \mathbb{B}_r$ with $r = (\frac{p^{-\frac{1}{k}} \tilde{r}_k}{2})^{\frac{1}{\theta-1}}$ and $p = \frac{d}{n\sqrt{\rho}}$, then, if $n$ is large enough such as $n \geq \Omega(\frac{\sqrt{d}}{\sqrt{\rho}})$, we have the following lower bound

$$\mathcal{M}(\mathcal{W}, \mathcal{P}, \mathcal{F}_k^\theta(\mathcal{P}, \tilde{r}^{(k)}), \rho) \geq \Omega \left( (\tilde{r}_k((\frac{\sqrt{d}}{\sqrt{\rho}n})^{\frac{k-1}{k}}))^{\frac{\theta}{\theta-1}} \right).$$

**Theorem 5.** For any $\theta, k \geq 2, \tilde{r}_k > 0$, denote $\mathcal{X} = \{\pm \frac{\tilde{r}_k}{2\sqrt{d}}\}^d$, and $\mathcal{W} = \mathbb{B}_r$ with $r = (\frac{\tilde{r}_k}{2})^{\frac{1}{\theta-1}}$, then, if $n \geq \Omega(\sqrt{d})$, we have the following lower bound

$$\mathcal{M}(\mathcal{W}, \mathcal{P}, \mathcal{F}_k^\theta(\mathcal{P}, \tilde{r}^{(k)}), \rho) \geq \Omega \left( (\frac{\tilde{r}_k}{\sqrt{n}})^{\frac{\theta}{\theta-1}} \right).$$

**Remark 5.** First, it is notable that although the upper bounds in Section 4 are for $(\epsilon, \delta)$-DP, we can easily extend to the $\rho$-zCDP case as we used the Gaussian mechanism and parallel theorem to guarantee DP, which also hold for zCDP (10). The only difference is replacing the term $O(\frac{\sqrt{\log \frac{1}{\delta}}}{\epsilon})$ by $O(\frac{1}{\sqrt{\rho}})$. Thus, from this side, combining with Theorem 4 and 5, we can see the upper bound is nearly optimal for $\rho$-zCDP in the general case if $\tilde{r}_{2k}$ (since $\widetilde{R}_{2k,n} \leqslant \tilde{r}_{2k}$) and $\tilde{r}_k$ are asymptotically the same. Secondly, in the Lipschitz case for $(\epsilon, \delta)$-DP, (2) proved the lower bound result via a reduction to the ERM problem for general convex loss. However, their reduction cannot be applied to our problem as their proof heavily relies on the $O(1)$-Lipschizt condition, which is not satisfied for our loss. For $\epsilon$-DP, (2) considered the empirical risk and used the packing argument for the lower bound, which cannot be applied to our problem as our loss is not constant Lipschitz. In our proof, we directly considered the population risk $F_P(w) = -\langle w, \mathbb{E}_P[x] \rangle + \frac{1}{\theta} \|w\|_2^\theta$ for some data distribution $P$ and used private Fano's lemma to prove the lower bound.

## 6 RELAX THE LIPSCHITZ ASSUMPTION

---

**Algorithm 6** Permuted Noisy Clipped Accelerated SGD for Heavy-Tailed DP SCO (PNCA-SGD)

---

**Input:** Data $S \in \mathcal{X}^n$, iteration number $T$, stepsize parameters $\{\eta_t\}_{t\in[T]}$, $\{\alpha_t\}_{t\in[T]}$ with $\alpha_1 = 1$, private paratemter $\epsilon, \delta$, initialization $w_0$.

1: Randomly permute the data and denote the permuted data as $\{x_1, \cdots, x_n\}$.
2: Initialize $w_0^{ag} = w_0$.
3: **for all** $t \in [T]$ **do**
4:     $w_t^{md} := (1 - \alpha_t) w_{t-1}^{ag} + \alpha_t w_{t-1}$.
5:     Draw new batch $\mathcal{B}_t$ (without replacement) of $n/T$ samples from $S$.
6:     $\widetilde{\nabla} F_t\left(w_t^{md}\right) := \text{ClippedMean}\left(\left\{\nabla f\left(w_t^{md}, x\right)\right\}_{x \in \mathcal{B}_t}; \frac{n}{T}; C\right) + \zeta_i$, where $\zeta_i \sim \mathcal{N}(0, \sigma^2 \mathbb{I}_d)$,

       $\sigma^2 = O(\frac{C^2 T \log \frac{1}{\delta}}{n^2 \epsilon^2})$ and $C = \widetilde{r}_k \left(\frac{\epsilon n}{\sqrt{d \log(1/\delta)}}\right)^{1/k}$.

7:     $w_t := \arg_{w \in \mathcal{W}} \left\{ \alpha_t \left\langle \tilde{\nabla} F_t\left(w_t^{md}\right), w \right\rangle + \frac{\eta_t}{2} \|w_{t-1} - w\|^2 \right\}$.
8:     $w_t^{ag} := \alpha_t w_t + (1 - \alpha_t) w_{t-1}^{ag}$.
9: **end for**
**Return** $w_T^{ag}$

---

In the previous section, we have considered the Lipschitz case and show that under the TNC, compared to that for the general convex loss, it is possible to get improved excess population risk that is independent of the Lipschitz constant. There are still two questions left: (1) Compared to the previous studies on DP-SCO with heavy-tailed gradient such as (41; 20), our above upper bounds still need the finite Lipschitz condition; (2) We can see our upper bounds depend on $\tilde{R}_{2k,n} \leqslant \tilde{r}_{2k}$ while the lower bounds only depend on $\tilde{r}_k$. Thus, there is a gap for the moment term. In this section, we aim to address these two issues. Specifically, we will show that even if the loss function is not Lipschitz, it is still possible to get the same upper bound as in the above section when $\epsilon$ is small enough. Moreover, we can improve the dependency from $\tilde{R}_{2k,n}$ to $\tilde{r}_k$.

Specifically, our main method, Algorithm 7, shares a similar idea as in Algorithm 5 with different parameters and base algorithm. Specifically, rather than using Algorithm 3, here we propose Algorithm 6 as our base algorithm, which is a shuffled, clipped, and private version of the accelerated SGD. Specifically, in step 1 we randomly shuffle the data for privacy amplification (15). Then, in each iteration, we clipped the gradients and added Gaussian noise to ensure DP. We can show that with some parameters, the output could achieve an upper bound similar to Theorem 1 even if the loss is not Lipschitz.

---

**Algorithm 7** Iterated PNCA-SGD $\left(w_0, n, \mathcal{W}, \bar{\theta}\right)$

---

**Input:** Dataset $S \in \mathcal{X}^n$, initial point $w_0 \in \mathcal{W}$, privacy parameter $\epsilon$ and $\delta$.

1: Partite the data $S$ into $k$ disjoint subsets $\{S_1, \cdots, S_k\}$, where $k = \lfloor (\log_{\bar{\theta}} 2) \cdot \log \log n \rfloor$, and for each $i \in [k], |S_i| = n_i = \left\lfloor 2^{i-1} n / (\log n)^{\log_{\bar{\theta}}^2 2} \right\rfloor$.
2: **for all** $t = 1, \cdots, k$ **do**
3:     Let $w_t = \text{PNCA-SGD}\left(w_{t-1}, \eta_t, n_t, \mathcal{W}\right)$, where the AC-SA runs on the $t$-th subset $S_i$. For $(\epsilon, \delta)$-DP, $\eta_t = \frac{4\eta}{t(t+1)}, \alpha_t = \frac{2}{t+2}$ and $R_l = \frac{R_{l-1}}{2}$.
4: **end for**
**Return** $w_k$

---

**Theorem 6.** *For any $\epsilon = O(\sqrt{\frac{\log n/\delta}{n}})$, and $0 < \delta < 1$, Algorithm 6 is $(\epsilon, \delta)$-DP. Moreover, under Assumption 1 and assume function $F$ is $\beta$-smooth with the diameter $D$ over $w \in \mathcal{W}$, then the output of Algorithm 6, by selecting the following $T$,*

$$T = \min\{\sqrt{\frac{\beta D}{\widetilde{r}_k}} \cdot (\frac{\epsilon n}{\sqrt{d \log(1/\delta)}})^{\frac{k-1}{2k}}, \sqrt{\frac{\beta D}{\widetilde{r}_k}} \cdot n^{1/4}\},$$

*we have*

$$\mathbb{E}F\left(w_T^{ag}\right) - F^* \leqslant O\left(\widetilde{r}_k D(\frac{1}{\sqrt{n}} + (\frac{\sqrt{d\log(1/\delta)}}{\epsilon n})^{\frac{k-1}{k}})\right).$$

Note that (29) also proposes a private accelerated SGD. However, their bound is sub-optimal (see the second row is Table 1). Here, we leverage privacy amplification via shuffling to reduce the noise added to each iteration. Thus, we can get the optimal rate here. We note that this is also the first result that can achieve the optimal rate for the general convex function without the Lipschitz assumption. Based on this result, we have the following theorem for Algorithm 7.

**Theorem 7.** *For any $\epsilon = O(\sqrt{\frac{\log n/\delta}{n}})$, and $0 < \delta < 1$, Algorithm 7 is $(\epsilon, \delta)$-DP. Moreover, under Assumption 1 and assume function $F$ is $\beta$-smooth, then we have*

$$\mathbb{E}F(\hat{w}_m) - F(w^*) \leqslant \tilde{O}\left(\frac{1}{\lambda^{\frac{1}{\theta-1}}}(\widetilde{r}_k(\frac{1}{\sqrt{n}} + (\frac{\sqrt{d}}{\epsilon n})^{\frac{k-1}{k}}))^{\frac{\theta}{\theta-1}}\right). \tag{3}$$

Compared with the results in the above section, we can see the result in Theorem 7 is in the expectation form, which is due to the noisy clipped gradient in Algorithm 6. Moreover, the constraint of $\epsilon = O(\sqrt{\frac{\log n/\delta}{n}})$ comes from the results of privacy amplification via shuffling (15). We leave these two assumptions to be relaxed for future research. Moreover, the improvement from $\widetilde{R}_{2k,n}$ to $\widetilde{r}_k$ is due to the different results between Theorem 6 and 1.

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

## A  OMITTED PROOFS

**Proof of Lemma 2.** Let $l \in \mathbb{N}$, and $n = 2^l$ and consider

$$\widehat{r}_n(X)^{(k)} = \frac{1}{n} \sup_w \left( \sum_{i=1}^{n/2} \|\nabla f(w, x_i)\|^k + \sum_{i=n/2+1}^{n} \|\nabla f(w, x_i)\|^k \right)$$

$$\leqslant \frac{1}{n} \left( \sup_w \sum_{i=1}^{n/2} \|\nabla f(w, x_i)\|^k + \sup_w \sum_{i=n/2+1}^{n} \|\nabla f(w, x_i)\|^k \right).$$

Taking expectations over the random draw of $X \sim \mathcal{D}^n$ and we have $\widehat{e}_n^{(k)} \leqslant \widehat{e}_{n/2}^{(k)}$. Thus, $\widetilde{R}_{k,n} \leqslant \widetilde{r}_k$. $\qquad\square$

**Proof of Theorem 1.  Privacy.** Since in each epoch of Algorithm 3 we use a disjoint dataset, it is sufficient for us to show each $w_i$ is $(\epsilon, \delta)$-DP. Recall the following lemma:

**Lemma 3.** *[Theorem 3.9 in ((17))] Assume the loss function $f(\cdot, x)$ is L-Lipschitz, $\lambda$-strongly convex and $\beta$-smooth with respect to $w \in \mathcal{W}$ for all $x$. Let $S_i$ and $S_i'$ be two samples of size $n_i$ differing in only a single element. Denote $w_i^t$ and $w'^t_i$ as the outputs of the projected stochastic gradient method on datasets $S_i$ and $S_i'$ respectively, then if $\eta \leqslant \frac{1}{\beta}$ we have*

$$\left\| w_i^t - w'^t_i \right\| \leqslant \frac{2L}{\lambda n_i}$$

We can see $F_i$ is $\lambda_i$-strongly convex and the gradient after clipping $\Pi_C(\nabla f(\cdot, x))$ is also $\alpha$-Lipschitz if $f(\cdot, x)$ is $\alpha$-smooth, which is due to that

$$\|\Pi_C(\nabla f(w, x)) - \Pi_C(\nabla f(w', x))\| \leqslant \|\nabla f(w, x) - \nabla f(w', x)\|_2 \leqslant \alpha \|w - w'\|_2. \qquad (4)$$

Thus, by a similar proof for Lemma 3, we can see that $\|\hat{w}_i - \hat{w}_i'\|_2 \leqslant \frac{2C_i}{n_i \lambda_i}$, where $\hat{w}_i'$ is the output of Algorithm 2 when the input is $S_i'$. Thus, by the Gaussian mechanism we can see it is $(\epsilon, \delta)$-DP.

**Excess risk:** We finish our proof through several parts. We first recall the following lemma.

**Lemma 4.** *[(16)] Assume $\mathrm{diam}_2(\mathcal{X}) \leqslant D$. Let $\mathcal{S} = (S_1, \ldots, S_n)$ where $S_1^n \overset{iid}{\sim} P$ and $f(w, x)$ is L-Lipschitz and $\lambda$-strongly convex for all $x \in \mathcal{X}$. Let $\hat{x} = \arg\min_{x \in \mathcal{X}} \bar{F}(w)$ be the empirical minimizer. For $0 < \beta \leqslant 1/n$, with probability at least $1 - \beta$*

$$F(\hat{x}) - F(x^\star) \leqslant \frac{cL^2 \log(n) \log(1/\beta)}{\lambda n} + \frac{cLD \sqrt{\log(1/\beta)}}{\sqrt{n}}.$$

**Theorem 8.** *We have the following bound for $\|w_T - \hat{w}\|^2$ for $T$ iterations:*

$$\|w_T - \hat{w}\|^2 \leqslant \exp\{-\frac{\lambda \eta T}{2}\} \|w_0 - \widehat{w}\|^2 + \frac{8\eta \hat{r}_n^2(x)}{\lambda} + 8\eta \lambda D^2 + \frac{20\widehat{B}^2}{\lambda^2}.$$

*Proof.* Detailly,

$$\left\| \tilde{\nabla} F_\lambda(w_t) \right\|^2 \leqslant 2 \left( \left\| \nabla \widehat{F}_\lambda(w_t) \right\|^2 + \|b_t\|^2 \right)$$

$$\leqslant 2 \left( 2\widehat{r}_n(X)^2 + 2\lambda^2 D^2 + \hat{B}^2 \right),$$

And also, by Young's inequality,

$$|\langle b_t, w_t - \hat{w} \rangle| \leqslant \frac{\widehat{B}^2}{\lambda} + \frac{\lambda}{4} \|w_t - \hat{w}\|^2.$$

Set $\widetilde{\nabla} F_\lambda(w_t) = \nabla \hat{F}_\lambda(w_t) + b_t = \frac{1}{n} \sum_{i=1}^{n} \Pi_C(\nabla f(w, x_i)) + \lambda(w - w_0)$ as the biased, noisy subgradients of the regularized empirical loss in Algorithm 3, with $N_t \sim \mathcal{N}(0, \sigma^2 \mathbf{I}_d)$ and $b_t =$

$\frac{1}{n}\sum_{i=1}^{n}\Pi_{C}\left(\nabla f\left(w_{t},x_{i}\right)\right)-\frac{1}{n}\sum_{i=1}^{n}\nabla f\left(w_{t},x_{i}\right)$. Denote $y_{t+1}=w_{t}-\eta\tilde{\nabla}F_{\lambda}\left(w_{t}\right)$, so that $w_{t+1}=\Pi_{\mathcal{W}}\left(y_{t+1}\right)$. For now, by strong convexity, we have

$$
\begin{aligned}
\widehat{F}_{\lambda}\left(w_{t}\right)-\widehat{F}_{\lambda}(\hat{w}) &\leqslant \left\langle\nabla\widehat{F}_{\lambda}\left(w_{t}\right),w_{t}-\hat{w}\right\rangle-\frac{\lambda}{2}\left\|w_{t}-\hat{w}\right\|^{2} \\
&=\left\langle\tilde{\nabla}F_{\lambda}\left(w_{t}\right),w_{t}-\hat{w}\right\rangle-\frac{\lambda}{2}\left\|w_{t}-\hat{w}\right\|^{2}+\left\langle\nabla\widehat{F}_{\lambda}\left(w_{t}\right)-\tilde{\nabla}F_{\lambda}\left(w_{t}\right),w_{t}-\hat{w}\right\rangle \\
&=\frac{1}{2\eta}\left(\left\|w_{t}-\hat{w}\right\|^{2}+\left\|w_{t}-y_{t+1}\right\|^{2}-\left\|y_{t+1}-\hat{w}\right\|^{2}\right)-\frac{\lambda}{2}\left\|w_{t}-\hat{w}\right\|^{2} \\
&\quad +\left\langle\nabla\widehat{F}_{\lambda}\left(w_{t}\right)-\tilde{\nabla}F_{\lambda}\left(w_{t}\right),w_{t}-\hat{w}\right\rangle \\
&=\frac{1}{2\eta}\left(\left\|w_{t}-\hat{w}\right\|^{2}(1-\lambda\eta)-\left\|y_{t+1}-\hat{w}\right\|^{2}\right)+\frac{\eta}{2}\left\|\tilde{\nabla}F_{\lambda}\left(w_{t}\right)\right\|^{2} \\
&\quad +\left\langle\nabla\widehat{F}_{\lambda}\left(w_{t}\right)-\tilde{\nabla}F_{\lambda}\left(w_{t}\right),w_{t}-\hat{w}\right\rangle \\
&\leqslant\frac{1}{2\eta}\left(\left\|w_{t}-\hat{w}\right\|^{2}(1-\lambda\eta)-\left\|w_{t+1}-\hat{w}\right\|^{2}\right)+\frac{\eta}{2}\left\|\tilde{\nabla}F_{\lambda}\left(w_{t}\right)\right\|^{2}-\left\langle b_{t},w_{t}-\hat{w}\right\rangle,
\end{aligned}
$$

where we used non-expansiveness of projection and the definition of $\tilde{\nabla}F_{\lambda}\left(w_{t}\right)$ in the last line. Now, re-arranging this inequality,

$$
\begin{aligned}
\left\|w_{t+1}-\hat{w}\right\|^{2} &\leqslant\left\|w_{t}-\hat{w}\right\|^{2}(1-\lambda\eta)+\eta^{2}\left\|\tilde{\nabla}F_{\lambda}(w_{t})\right\|^{2}-2\eta\langle b_{t},w_{t}-w\rangle-2\eta(\widehat{F}_{\lambda}(w_{t})-\widehat{F}_{\lambda}(\hat{w})) \\
&\leqslant\left\|w_{t}-\hat{w}\right\|^{2}(1-\lambda\eta)+\eta^{2}\left\|\tilde{\nabla}F_{\lambda}(w_{t})\right\|^{2}-2\eta\langle b_{t},w_{t}-w\rangle \\
&\leqslant\left\|w_{t}-\hat{w}\right\|^{2}(1-\frac{\lambda\eta}{2})+\eta^{2}\cdot2(2\hat{r}_{n}^{2}(x)+2\lambda^{2}D^{2}+\widehat{B}^{2})+\frac{2\eta\widehat{B}^{2}}{\lambda} \\
&\leqslant\left\|w_{t}-\hat{w}\right\|^{2}(1-\frac{\lambda\eta}{2})+4\eta^{2}(\hat{r}_{n}^{2}(x)+\lambda^{2}D^{2}+\widehat{B}^{2})+\frac{2\eta\widehat{B}^{2}}{\lambda},
\end{aligned}
$$

where $\widehat{B}$ is defined as below,

$$
\widehat{B}=\sup_{t\in T}\left\|b_{t}\right\|\leqslant\frac{\widehat{r}_{n}(X)^{(k)}}{(k-1)C^{k-1}}.
$$

Thus, iterating the above equation, we get

$$
\begin{aligned}
\left\|w_{T}-\hat{w}\right\|^{2} &\leqslant(1-\frac{\lambda\eta}{2})^{T}\left\|w_{0}-\hat{w}\right\|^{2}+(4\eta^{2}(\hat{r}_{n}^{2}(x)+\lambda^{2}D^{2}+\widehat{B}^{2})+\frac{2\eta\widehat{B}^{2}}{\lambda})\sum_{t=1}^{T-1}(1-\frac{\lambda\eta}{2})^{t} \\
&\leqslant(1-\frac{\lambda\eta}{2})^{T}\left\|w_{0}-\hat{w}\right\|^{2}+(4\eta^{2}(\hat{r}_{n}^{2}(x)+\lambda^{2}D^{2}+\widehat{B}^{2})+\frac{2\eta\widehat{B}^{2}}{\lambda})\frac{2}{\lambda\eta} \\
&=(1-\frac{\lambda\eta}{2})^{T}\left\|w_{0}-\hat{w}\right\|^{2}+\frac{8\eta}{\lambda}(\hat{r}_{n}^{2}(x)+\lambda^{2}D^{2}+\widehat{B}^{2})+\frac{4\widehat{B}^{2}}{\lambda^{2}} \\
&\leqslant\exp\{-\frac{\lambda\eta T}{2}\}\left\|w_{0}-\widehat{w}\right\|^{2}+\frac{8\eta\hat{r}_{n}^{2}(x)}{\lambda}+8\eta\lambda D^{2}+\frac{8\eta\widehat{B}^{2}}{\lambda}+\frac{4\widehat{B}^{2}}{\lambda^{2}} \\
&\leqslant\exp\{-\frac{\lambda\eta T}{2}\}\left\|w_{0}-\widehat{w}\right\|^{2}+\frac{8\eta\hat{r}_{n}^{2}(x)}{\lambda}+8\eta\lambda D^{2}+\frac{20\widehat{B}^{2}}{\lambda^{2}}.
\end{aligned}
$$

The last inequality holds due to the assumption that $\eta\leqslant\frac{2}{\lambda}$. $\qquad\square$

**Theorem 9.** *We have the following bound for $f(w_{l})-f(\hat{w}_{l})$:*

$$
F(w_{l})-F(\hat{w}_{l})\leqslant\widetilde{O}\left(\frac{\widetilde{r}_{2k,n_{l}}}{\widetilde{R}_{2k,n}}\cdot\frac{DL_{f}}{\sqrt{n}}\right).
$$

*Proof.* Firstly, the choise of $D_i$ ensures that $\hat{w}_i \in \mathcal{W}_i$.

Then by the above lemma, and choosing specific $T_i$,

$$\|w_i - \hat{w}_i\|^2 \leqslant \exp\{-\frac{\lambda_i \eta_i T_i}{2}\} \|w_{i-1} - \hat{w}_i\|^2 + \frac{8\eta_i \hat{r}_{n_i}^2 (B_i)^{(2)}}{\lambda_i} + 8\eta_i \lambda_i D_i^2 + \frac{20\hat{r}_{n_i}^2 (B_i)^{(2)}}{\lambda_i^2 (k-1) C_i^{k-1}}.$$

$$\|w_i - \hat{w}_i\|^2 \lesssim \frac{\eta_i}{\lambda_i} L_f^2 + \frac{\widetilde{r}_{n_i}^{(2k)}}{\lambda_i^2 C_i^{2k-2} 4^i} \lesssim \frac{\eta^2 n}{16^i 4^i} (L_f^2 + \frac{n \widetilde{r}_{n_i}^{(2k)}}{C_i^{2k-2} 4^i}). \tag{5}$$

Then by setting $L = \sup_{w \in \mathcal{W}} \|\nabla F(w)\| \leqslant r$. Therefore,

$$F(w_l) - F(\hat{w}_l) \leqslant \sqrt{\|w_l - \hat{w}_l\|^2}$$

$$\leqslant L\sqrt{\eta_l^2 (L_f^2 + \frac{\widetilde{e}_{n_i}^{(2k)}}{C_l^{2k-2} 4^i})}$$

$$\lesssim L\frac{\eta}{n^2} (L_f + \frac{\widetilde{r}_{2k}^k}{C_l^{k-1}})$$

$$\lesssim L\frac{\eta}{n^2} (L_f + \frac{\widetilde{r}_{2k}^k}{C_l^{k-1}})$$

$$\leqslant L\frac{\eta}{n^2} \left( L_f + \widetilde{r}_{2k} (\frac{\sqrt{d}}{\epsilon})^{\frac{k-1}{k}} \right).$$

We know that $\xi_i \sim \mathcal{N}(0, \sigma_i^2)$ and $\xi$ is sub-Gaussian, thus, we can derive that

$$\mathbb{P}\{\|\xi_i\| \geqslant t\sqrt{d}\} \leqslant 2\exp\{-\frac{t^2}{16\sigma_i^2}\}.$$

Here there shall be some confusion about the lower index, where $k$ is equivalent to $l$ as above, not the original $k$ here. Therefore, with probability $1 - \beta$, $\|\xi_i\| \leqslant 4\sqrt{d}\sigma_i \log(4/\beta)$. Thus, due to the choice of $\eta$, we have

$$F(w_l) - F(\hat{w}_l) \leqslant 4L_f\sqrt{d}\sigma_l \log(4/\beta) = 4L_f\sqrt{d}\log(4/\beta) \frac{8C_l\sqrt{\log(1/\delta)}}{n_l \lambda_l \epsilon}$$

$$= 32L_f\sqrt{d}\log(4/\beta) \frac{C_l \eta_l n_l^{p-1}}{\epsilon}$$

$$= 32L_f\sqrt{d}\log(4/\beta)\widetilde{r}_{2k,n_l} \left( \frac{\epsilon n_l}{\sqrt{d\log(n)}} \right)^{\frac{1}{k}} \frac{\eta}{4^l} \frac{n^{p-1}}{(2^l)^{p-1}} \frac{1}{\epsilon}$$

$$\leqslant \frac{\widetilde{r}_{2k,n_l}}{\widetilde{R}_{2k,n}} \cdot \frac{32DL\log(1/\beta)}{\sqrt{n}\log^{p+\frac{5}{2}} n}.$$

$\square$

Finally, we reach the upper bound for $F(w_l) - F(w^*)$:

**Theorem 10.** *Finally, we reach the upper bound for $F(w_l) - F(w^*)$:*

$$F(w_l) - F(w^*) \lesssim \widetilde{R}_{2k,n} D(\frac{1}{\sqrt{n}} + (\frac{\sqrt{d\log n}}{\epsilon n})^{\frac{k-1}{k}}) + \frac{D\sqrt{\log(1/\beta)}}{2^{p+1}\sqrt{n}}.$$

*Proof.* Rewrite this term into summation of their differences,

$$F(w_l) - F(w^*) = \sum_{i=1}^{l} [f(\hat{w}_i) - f(\hat{w}_{i-1})] + [f(w_l) - f(\hat{w}_l)],$$

By lemma 4,

$$F(\hat{w}_i) - F(\hat{w}_{i-1}) \leqslant \frac{cL^2 \log n_i \log(2/\beta)}{\lambda_i n_i} + \frac{cLD_i\sqrt{\log(2/\beta)}}{\sqrt{n_i}} + \frac{\lambda_i}{2}\|w_{i-1} - \hat{w}_{i-1}\|^2.$$

For $\|w_i - \widehat{w}_i\|^2 \leqslant \frac{\eta_i}{\lambda_i}L_f^2 + \frac{\widetilde{r}_{n_i}^{(2k)}}{\lambda_i^2 C_i^{2k-2} 4^i} \leqslant O\left(\frac{\eta^2 n}{16^i 4^i}(L_f^2 + \frac{n\widetilde{r}_{n_i}^{(2k)}}{C_i^{2k-2} 4^i})\right)$, then summing over $i$ from 1 to $l$, we have with probability at least $1 - \beta$, for some constant $C_0$

$$f(w_l) - f(w^*)$$

$$\leqslant C_0 \sum_{i=1}^{l}\left\{\lambda_i\|\widehat{w}_{i-1} - w_{i-1}\|^2 + \frac{cL_f^2 \log n_i \log(1/\beta)}{\lambda_i n_i} + \frac{cL_f D_i\sqrt{\log(1/\beta)}}{\sqrt{n_i}}\right\}$$

$$\leqslant \lambda_1\|\widehat{w}_0 - w_0\|^2 + \sum_{i=2}^{l}\lambda_i\|\widehat{w}_{i-1} - w_{i-1}\|^2 + \sum_{i=1}^{l}\frac{L_f^2 \log n_i \log(1/\beta)}{\lambda_i n_i} + \sum_{i=1}^{l}\frac{L_f D_i\sqrt{\log(1/\beta)}}{\sqrt{n_i}}$$

$$\leqslant \frac{D^2}{\eta n^{2p}} + \sum_{i=2}^{l}\lambda_i\left[\eta_i^2 n_i^p L_f^2 + \frac{\eta_i^2 n_i^{2p}\widetilde{e}_{n_i}^{(2k)}}{C_i^{2k-2}}\right] + \sum_{i=1}^{l}\frac{L_f^2(\log n - \log 2^i)\log(1/\beta)}{n_i}\eta_i n_i^p + \sum_{i=1}^{l}L_f^2\eta_i n_i^{p-\frac{1}{2}}\sqrt{\log(1/\beta)}$$

$$\leqslant \frac{D^2}{\eta n^{2p}} + \sum_{i=2}^{l}\left[\eta_i^2 L_f^2 + \frac{\eta_i n_i^p\widetilde{e}_{n_i}^{(2k)}}{C_i^{2k-2}}\right] + \sum_{i=1}^{l}L_f^2\eta_i n_i^{p-1}(\log n - \log 2^i)\log(1/\beta) + \sum_{i=1}^{l}L_f^2\eta_i n_i^{p-\frac{1}{2}}\sqrt{\log(1/\beta)}$$

$$\leqslant \frac{D^2}{\eta n^{2p}} + \eta\left(L_f^2 + \widetilde{R}_{2k,n}n^p(\frac{d\log n}{\epsilon^2 n^2})^{\frac{k-1}{k}}\right) + L_f^2\eta n^{p-1}\log(1/\beta)\sum_{i=1}^{l}\frac{(\log n - i)}{4^i \cdot (2^{p-1})^i} + L_f^2\eta n^{p-\frac{1}{2}}\sqrt{\log(1/\beta)}\sum_{i=1}^{l}(\frac{1}{2^{p+\frac{3}{2}}})^i$$

$$\leqslant \frac{D^2}{\eta n^{2p}} + \eta\left(L_f^2 + \widetilde{R}_{2k,n}n^p(\frac{d\log n}{\epsilon^2 n^2})^{\frac{k-1}{k}}\right)$$

$$+ L_f^2\eta n^{p-1}\log(1/\beta)\left(\frac{\log n}{2^{p+1}} + \frac{1}{2^{p+1}\log^p n}\right) + L_f^2\eta n^{p-\frac{1}{2}}\sqrt{\log(1/\beta)}\frac{1 - \frac{1}{n^{p+\frac{3}{2}}}}{2^{p+\frac{3}{2}} - 1}$$

$$\leqslant \frac{D^2}{\eta n^{2p}} + \eta(L_f^2 + \widetilde{R}_{2k,n}n^p(\frac{d\log n}{\epsilon^2 n^2})^{\frac{k-1}{k}}) + L_f^2\eta n^{p-1}\log(1/\beta)\log n \cdot 2^{-(p+1)} + L_f^2\eta n^{p-\frac{1}{2}}\sqrt{\log(1/\beta)} \cdot 2^{-(p+\frac{3}{2})}.$$

Assume that $\exists p$ s.t. $L_f \leqslant O\left(n^{p/2}\widetilde{R}_{2k,n}(\frac{1}{\sqrt{n}} + (\frac{\sqrt{d\log n}}{\epsilon n})^{\frac{k-1}{k}})\right)$ and take $\eta = \frac{D}{n^{\frac{p}{2}}}\min\{\frac{1}{L_f}, \frac{1}{\widetilde{R}_{2k,n}n^{\frac{p+1}{2}}}(\frac{\epsilon n}{\sqrt{d\log n}})^{\frac{k-1}{k}}, \frac{1}{n^{\frac{p-1}{2}}L_f^2\sqrt{\log n\log(1/\beta)}}\}$, then the above can be reduced to

$$f(w_l) - f(w^*) \leqslant O\left(\widetilde{R}_{2k,n}D(\frac{1}{\sqrt{n}} + (\frac{\sqrt{d\log n}}{\epsilon n})^{\frac{k-1}{k}}) + \frac{D\sqrt{\log(1/\beta)}}{2^{p+1}\sqrt{n}}\right),$$

which holds with probability at least $1 - \beta$.

$\square$

$\square$

**Proof of Thorem 2.**

**Theorem 11.** *Assume that loss function $F(\cdot)$ is $(\theta, \lambda)$-TNC and $f(\cdot, x)$ is convex, $\alpha$-smooth and $L_f$-Lipschitz for each $x$. Then algorithm 4 is $(\epsilon, \delta)$-DP based on different stepsizes $\{\gamma_k\}_{k=1}^m$ and noises if $\gamma_k \leqslant \frac{1}{\alpha}$. Then for sufficiently large $n$ and $(\epsilon, \delta)$-DP, with probability at least $1 - \beta$, we have*

$$F(\hat{w}_m) - F(w^*) \leqslant O\left(\frac{1}{\lambda^{\frac{1}{\theta-1}}} \cdot \left(\widetilde{R}_{2k,n}(\frac{\sqrt{\log n}}{\sqrt{n}} + \left(\frac{\sqrt{d\log^3 n}}{\epsilon n}\right)^{\frac{k-1}{k}}) + \frac{\sqrt{\log n\log(1/\beta)}}{2^{p+1}\sqrt{n}}\right)^{\frac{\theta}{\theta-1}}\right).$$

*Proof.* The guarantee of $(\epsilon, \delta)$-DP is just followed by Theorem 1.

For simplicity, we denote $a(n) = O\left(\widetilde{R}_{2k,n}(\frac{1}{\sqrt{n}} + (\frac{\sqrt{d\log n}}{\epsilon n})^{\frac{k-1}{k}}) + \frac{\sqrt{\log(1/\beta)}}{2^{p+1}\sqrt{n}}\right)$. We set $\mu_0 = 2R_0^{1-\theta}a(n_0)$, $\mu_k = 2^{(\theta-1)k}\mu_0$ and $R_k = \frac{R_0}{2^k}$, where $k = 1, \cdots, m$.

Then we have $\mu_k \cdot R_k^\theta = 2^{-k}\mu_0 R_0^\theta$. We can also assume that $\lambda \leqslant \frac{L}{R_0^{\theta-1}}$, otherwise we can set $\lambda = \frac{L}{R_0^{\theta-1}}$, which makes TNC still hold. Recall that $m = \left\lfloor \frac{1}{2}\log_2\frac{2n}{\log_2 n}\right\rfloor - 1$, when $n \geq 256$, it follows that

$$0 < \frac{1}{2}\log_2\frac{2n}{\log_2 n} - 2 \leqslant m \leqslant \frac{1}{2}\log_2\frac{2n}{\log_2 n} - 1 \leqslant \frac{1}{2}\log_2 n.$$

Thus, we have $2^m \geq \frac{1}{4}\sqrt{\frac{2n}{\log_2 n}}$.(if we pick specific $m$ such that $2^m \geq \frac{1}{4}\sqrt{\frac{2n}{\log_2 n}} \cdot \frac{1}{\log n_0 \sqrt{\log(1/\beta)}}$ )

Thus

$$\mu_m = 2^{(\theta-1)m}\mu_0 \geq 2^m\mu_0$$

$$\geq \frac{1}{4}\sqrt{\frac{2n}{\log_2 n}}\frac{1}{\log n_0\sqrt{\log(1/\beta)}} \cdot 2 \cdot R_0^{1-\theta}a(n_0)$$

$$= \frac{5 \cdot R_0^{1-\theta}}{\log n_0\sqrt{\log(1/\beta)}}\sqrt{\frac{2n}{\log_2 n}}\left(\widetilde{R}_{2k,n_0}(\frac{1}{\sqrt{n_0}} + (\frac{\sqrt{d\log n_0}}{\epsilon n_0})^{\frac{k-1}{k}}) + \frac{\sqrt{\log(1/\beta)}}{2^{p+1}\sqrt{n_0}}\right)$$

$$\geq 5 \cdot \widetilde{R}_{2k,n_0}R_0^{1-\theta}\sqrt{\frac{2n}{\log_2 n}}\left(\frac{1}{\sqrt{\frac{2n}{\log_2 2n - \log_2\log_2 n - 4}}}\right)$$

$$= 5 \cdot \widetilde{R}_{2k,n_0}R_0^{1-\theta}\sqrt{\frac{\log_2 2n - \log_2\log_2 n - 4}{\log_2 n}} \cdot \log n_0\sqrt{\log(1/\beta)}$$

$$\geq \widetilde{R}_{2k,n_0}R_0^{1-\theta}\left(\text{Since } 5 \cdot \sqrt{\frac{\log_2 2n - \log_2\log_2 n - 4}{\log_2 n}} \geq 1 \text{ when } n \geq 256\right)$$

$$\geq \lambda(\text{ By assumption }).$$

where the third inequality is given by throwing away the $(\frac{\sqrt{d\log n_0}}{\epsilon n_0})^{\frac{k-1}{k}}$ and $\frac{\sqrt{\log(1/\beta)}}{2^{p+1}\sqrt{n_0}}$ term and substituting $m$ in term $\frac{1}{\sqrt{\frac{n}{m}}}$ with $\frac{1}{2}\log_2\frac{2n}{\log_2 n} - 2$. Below, we consider the following two cases.

**Case 1** If $\lambda \geq \mu_0$, then $\mu_0 \leqslant \lambda \leqslant \mu_m$. We have the following lemma.

**Lemma 5.** *Let $k^*$ satisfies $\mu_{k^*} \leqslant \lambda \leqslant 2^{\theta-1}\mu_{k^*}$, then for any $1 \leqslant k \leqslant k^*$, the points $\{\hat{w}_k\}_{k=1}^m$ generated by Algorithm 4 satisfy*

$$\|\hat{w}_{k-1} - w^*\|_2 \leqslant R_{k-1} = 2^{-(k-1)} \cdot R_0, \tag{6}$$

$$F(\hat{w}_k) - F(w^*) \leqslant \mu_k R_k^\theta = 2^{-k}\mu_0 R_0^\theta. \tag{7}$$

*Moreover, for $k \geq k^*$, we have*

$$F(\hat{w}_k) - F(\hat{w}_{k^*}) \leqslant \mu_{k^*}R_{k^*}^\theta. \tag{8}$$

*Proof.* We prove (6), (7) by induction. Note that (6) holds for $k = 1$. Assume (6) is true for some $k > 1$, then we have

$$F(\hat{w}_k) - F(w^*) \leqslant R_{k-1} \cdot \left(\widetilde{R}_{2k,n_0}(\frac{1}{\sqrt{n_0}} + (\frac{\sqrt{d\log n_0}}{\epsilon n_0})^{\frac{k-1}{k}}) + \frac{\sqrt{\log(1/\beta)}}{2^{p+1}\sqrt{n_0}}\right)$$

$$= R_{k-1}a(n_0)$$

$$= \frac{1}{2}\mu_k 2^{(1-\theta)k}R_0^{\theta-1}R_{k-1}$$

$$= \mu_k R_k^\theta$$

Which is (7). By the definition of TNC, we have

$$\|\hat{w}_k - w^*\|_2^\theta \leqslant \frac{1}{\lambda} \left( F\left(\hat{w}_k\right) - F\left(w^*\right) \right)$$
$$\leqslant \frac{F\left(\hat{w}_k\right) - F\left(w^*\right)}{\mu_{k^*}}$$
$$\leqslant \frac{\mu_k R_k^\theta}{\mu_{k^*}} \leqslant R_k^\theta$$

Thus (6) is true for $k+1$. Now we prove (8). Referring to Theorem 1 , we know that

$$F\left(\hat{w}_k\right) - F\left(\hat{w}_{k-1}\right) \leqslant R_{k-1} \cdot a\left(n_0\right)$$
$$= 2^{k^*-k} R_{k^*-1} a\left(n_0\right)$$
$$= 2^{k^*-k} \mu_{k^*} R_{k^*}^\theta$$
$$= \mu_k R_k^\theta$$

Thus, for $k > k^*$,

$$F\left(\hat{w}_k\right) - F\left(\hat{w}_{k^*}\right) = \sum_{j=k^*+1}^{k} \left( F\left(\hat{w}_j\right) - F\left(\hat{w}_{j-1}\right) \right)$$
$$\leqslant \sum_{j=k^*+1}^{k} 2^{k^*-j} \mu_{k^*} R_{k^*}^\theta$$
$$= \left( 1 - 2^{k^*-k} \right) \mu_{k^*} R_{k^*}^\theta$$
$$\leqslant \mu_{k^*} R_{k^*}^\theta$$

$\square$

Here completes the proof of the lemma. Now we proceed to prove Theorem 1 in this case.

$$F\left(\hat{w}_m\right) - F\left(w^*\right) = \left( F\left(\hat{w}_m\right) - F\left(\hat{w}_{k^*}\right) \right) + \left( F\left(\hat{w}_{k^*}\right) - F\left(w^*\right) \right)$$
$$\leqslant 2\mu_{k^*} R_{k^*}^\theta$$
$$\leqslant 4 \left( \frac{\mu_{k^*}}{\lambda} \right)^{\frac{1}{\theta-1}} \mu_{k^*} R_{k^*}^\theta \quad \left( \text{Since } \left( \frac{\mu_{k^*}}{\lambda} \right)^{\frac{1}{\theta-1}} \geq \frac{1}{2} \right)$$
$$= 4 \left( \frac{2^{(\theta-1)k^*} \mu_0}{\lambda} \right)^{\frac{1}{\theta-1}} \mu_{k^*} R_{k^*}^\theta$$
$$= 4 \left( 2^{k^*} \mu_{k^*} R_{k^*}^\theta \mu_0^{\frac{1}{\theta-1}} \left( \frac{1}{\lambda} \right)^{\frac{1}{\theta-1}} \right)$$
$$= 4 \left( \mu_0 R_0^\theta \mu_0^{\frac{1}{\theta-1}} \left( \frac{1}{\lambda} \right)^{\frac{1}{\theta-1}} \right)$$
$$= 4 \left( R_0^\theta \mu_0^{\frac{\theta}{\theta-1}} \left( \frac{1}{\lambda} \right)^{\frac{1}{\theta-1}} \right)$$
$$= 4 \cdot \left( (2 \cdot a\left(n_0\right))^{\frac{\theta}{\theta-1}} \left( \frac{1}{\lambda} \right)^{\frac{1}{\theta-1}} \right)$$
$$= 4 \cdot \left( \frac{1}{\lambda} \right)^{\frac{1}{\theta-1}} \cdot 2 \left( \widetilde{R}_{2k,n_0} \left( \frac{1}{\sqrt{n_0}} + \left( \frac{\sqrt{d \log n_0}}{\epsilon n_0} \right)^{\frac{k-1}{k}} \right) + \frac{\sqrt{\log(1/\beta)}}{2^{p+1}\sqrt{n_0}} \right)^{\frac{\theta}{\theta-1}}$$

where $m = O\left(\log_2 n\right)$ ( Recall that $m \leqslant \frac{1}{2} \log_2 n$).

**Case 2** If $\lambda < \mu_0$, then

$$F\left(\hat{w}_1\right) - F\left(w^*\right) \leqslant R_0 a\left(n_0\right)$$

$$= \left(\frac{2}{\mu_0}\right)^{\frac{1}{\theta-1}} \cdot a\left(n_0\right)^{\frac{\theta}{\theta-1}}$$

$$< \left(\frac{2}{\lambda}\right)^{\frac{1}{\theta-1}} \cdot a\left(n_0\right)^{\frac{\theta}{\theta-1}}$$

Also, we have

$$F\left(\hat{w}_m\right) - F\left(\hat{w}_1\right) = \sum_{j=2}^{m} \left(F\left(\hat{w}_j\right) - F\left(\hat{w}_{j-1}\right)\right)$$

$$\leqslant \sum_{j=2}^{m} R_{j-1} \cdot a\left(n_0\right)$$

$$= \sum_{j=2}^{m} 2^{-(j-1)} R_0 \cdot a\left(n_0\right)$$

$$= \left(1 - (1/2)^{m-1}\right) R_0 \cdot a\left(n_0\right) < R_0 \cdot a\left(n_0\right)$$

By a similar argument process as in Case 1, we have

$$F\left(\hat{w}_m\right) - F\left(w^*\right) = \left(F\left(\hat{w}_m\right) - F\left(\hat{w}_1\right)\right) + \left(F\left(\hat{w}_1\right) - F\left(w^*\right)\right)$$

$$\leqslant 2R_0 a\left(n_0\right) \leqslant 2\left(\frac{2}{\lambda}\right)^{\frac{1}{\theta-1}} \cdot a\left(n_0\right)^{\frac{\theta}{\theta-1}}$$

$$= 2 \cdot \left(\frac{2}{\lambda}\right)^{\frac{1}{\theta-1}} \cdot \left(\widetilde{R}_{2k,n_0}(\frac{1}{\sqrt{n_0}} + (\frac{\sqrt{d\log n_0}}{\epsilon n_0})^{\frac{k-1}{k}}) + \frac{\sqrt{\log(1/\beta)}}{2^{p+1}\sqrt{n_0}}\right)^{\frac{\theta}{\theta-1}}$$

Combining the two cases, we conclude that with probability at least $1 - \beta$,

$$F\left(\hat{w}_m\right) - F\left(w^*\right) \leqslant O\left(\frac{1}{\lambda^{\frac{1}{\theta-1}}} \cdot \left(\widetilde{R}_{2k,n}(\frac{\sqrt{\log n}}{\sqrt{n}} + \left(\frac{\sqrt{d\log^3 n}}{\epsilon n}\right)^{\frac{k-1}{k}}) + \frac{\sqrt{\log n \log(1/\beta)}}{2^{p+1}\sqrt{n}}\right)^{\frac{\theta}{\theta-1}}\right).$$

$$\square$$

$$\square$$

**Proof of Theorem 3.** *Proof.* The guarantee of $(\epsilon, \delta)$-DP is just followed by Theorem 1 and the parallel theorem of Differential Privacy. In the following we focus on the utility.

Since $k = \lfloor (\log \log_{\bar{\theta}} 2) \cdot \log \log n \rfloor$, then $k \leqslant (\log_{\bar{\theta}} 2) \cdot \log \log n$, namely $2^k \leqslant (\log n)^{\log 2}$ and $\frac{2^k - 1}{(\log n)^{\log_{\theta} 2}} \leqslant 1$. Observe that the total sample number used in the algorithm is $\sum_{i=1}^{k} n_i \leqslant \sum_{i=1}^{k} \frac{2^{i-1} n}{(\log n)^{\log_{\bar{\theta}} 2}} = \frac{(2^k - 1)n}{(\log n)^{\log_{\bar{\theta}} 2}} \leqslant n$.

For the output of phase $i$, denote $\Delta_i = F\left(w_i\right) - F\left(w^*\right)$, and let $D_i^{\theta} = \|w_i - w^*\|_2^{\theta}$. The assumption of TNC implies that $F\left(w_i\right) - F\left(w^*\right) \geq \lambda \|w_i - w^*\|_2^{\theta}$, which is $F\left(w_i\right) - F\left(w^*\right) \geq \lambda \|w_i - w^*\|_2^{\theta}$ when we take expectations at both sides, namely

$$\Delta_i \geq \lambda D_i^{\theta}. \tag{9}$$

Thus, we have

$$
\Delta_i \leqslant c\widetilde{R}_{2k,n}D_{i-1}\left(\frac{1}{\sqrt{n_i}} + \left(\frac{\sqrt{d\log n_i}}{\epsilon n_i}\right)^{\frac{k-1}{k}}\right) + \frac{cD_{i-1}\sqrt{\log(1/\beta)}}{2^{p+1}\sqrt{n_i}}
$$
$$
\overset{(9)}{\leqslant} \left(\frac{\Delta_{i-1}}{\lambda}\right)^{\frac{1}{\theta}}\left(c\widetilde{R}_{2k,n}\left(\frac{1}{\sqrt{n_i}} + \left(\frac{\sqrt{d\log n_i}}{\epsilon n_i}\right)^{\frac{k-1}{k}}\right) + \frac{c\sqrt{\log(1/\beta)}}{2^{p+1}\sqrt{n_i}}\right), \tag{10}
$$

where the first inequality comes from Theorem 1 and the second inequality uses (9). Denote $E_i = \frac{c^\theta}{\lambda}\left(\widetilde{R}_{2k,n}\left(\frac{1}{\sqrt{n_i}} + \left(\frac{\sqrt{d\log n_i}}{\epsilon n_i}\right)^{\frac{k-1}{k}}\right) + \frac{\sqrt{\log(1/\beta)}}{2^{p+1}\sqrt{n_i}}\right)^\theta$. Then (10) can be simplified as

$$
\Delta_i \leqslant (\Delta_{i-1}E_i)^{\frac{1}{\theta}}. \tag{11}
$$

Notice that $n_i/n_{i-1} = 2$, then $\frac{E_{i-1}}{E_i} \leqslant \left(\frac{n_i}{n_{i-1}}\right)^\theta = 2^\theta$, namely:

$$
E_i \geq 2^{-\theta}E_{i-1}. \tag{12}
$$

Then we can rearrange the above inequality as

$$
\frac{\Delta_i}{E_i^{\frac{1}{\theta-1}}} \leqslant \frac{(\Delta_{i-1}E_i)^{\frac{1}{\theta}}}{E_i^{\frac{1}{\theta-1}}} \leqslant 2^{\frac{1}{\theta-1}}\left(\frac{\Delta_{i-1}}{E_{i-1}^{\frac{1}{\theta-1}}}\right)^{\frac{1}{\theta}}, \tag{13}
$$

where the first inequality uses (11) and the second inequality applies (12).

It can be verified that (13) is equivalent to

$$
\frac{\Delta_i}{2^{\frac{\theta}{(\theta-1)^2}}E_i^{\frac{1}{\theta-1}}} \leqslant \left(\frac{\Delta_{i-1}}{2^{\frac{\theta}{(\theta-1)^2}}E_{i-1}^{\frac{1}{\theta-1}}}\right)^{\frac{1}{\theta}} \leqslant \left(\frac{\Delta_1}{2^{\frac{\theta}{(\theta-1)^2}}E_1^{\frac{1}{\theta-1}}}\right)^{\frac{1}{\theta^{i-1}}}.
$$

According to Lemma 1, $\Delta_1 \leqslant \left(L^\theta\lambda^{-1}\right)^{\frac{1}{\theta-1}}$. Also observe that

$$
E_1 = \frac{c^\theta}{\lambda}\left(\widetilde{R}_{2k,n}\left(\frac{1}{\sqrt{n_1}} + \left(\frac{\sqrt{d\log n_1}}{\epsilon n_1}\right)^{\frac{k-1}{k}}\right) + \frac{\sqrt{\log(1/\beta)}}{2^{p+1}\sqrt{n_1}}\right)^\theta \geq \frac{c^\theta\widetilde{R}_{2k,n}^\theta}{\lambda}\frac{1}{\left(\sqrt{n_1}\right)^\theta} \geq \frac{c^\theta\widetilde{R}_{2k,n}^\theta}{\lambda}\frac{1}{n^\theta}.
$$

Let $c_1 = c^{\frac{\theta}{\theta-1}}2^{\frac{\theta}{(\theta-1)^2}}$, then $\frac{\Delta_1}{2^{\frac{\theta}{(\theta-1)^2}}E_1^{\frac{1}{\theta-1}}} \leqslant \frac{n^{\frac{\theta}{\theta-1}}}{c_1}$, which implies that for $l = \lfloor(\log_{\bar\theta}2)\cdot\log\log n\rfloor$,

$$
\frac{\Delta_l}{2^{\frac{\theta}{(\theta-1)^2}}E_l^{\frac{1}{\theta-1}}} \leqslant \left(\frac{n^{\frac{\theta}{\theta-1}}}{c_1}\right)^{\frac{1}{\theta^{l-1}}}.
$$

Let $C_1 = 2^{\frac{\theta^3}{\theta-1}+\theta^2|\log c_1|}$. In the following we prove that

$$
\left(\frac{n^{\frac{\theta}{\theta-1}}}{c_1}\right)^{\frac{1}{\theta^{l-1}}} \leqslant C_1.
$$

Since $l+1 \geq (\log_{\bar\theta}2)\log\log n \geq (\log_\theta2)\log\log n$, it follows that

$$
(l-1)\log\theta + \log\log C_1 \geq \log\left(\frac{\theta}{\theta-1} + |\log c_1|\right) + \log\log n,
$$

which indicates

$$
\left(\frac{\theta}{\theta-1} + |\log c_1|\right)\log n \leqslant \theta^{l-1}\log C_1.
$$

Thus we have $\frac{\theta}{\theta-1} \log n - \log c_1 \leqslant \theta^{l-1} \log C_1$, which is equivalent to our object $\left(\frac{n^{\frac{\theta}{\theta-1}}}{c_1}\right)^{\frac{1}{\theta^{k-1}}} \leqslant$ $C_1$. Now we know

$$\frac{\Delta_l}{2^{\frac{\theta^2}{(\theta-1)^2}} E_l^{\frac{1}{\theta-1}}} \leqslant \left(\frac{n^{\frac{\theta}{\theta-1}}}{c_1}\right)^{\frac{1}{\theta^{l-1}}} \leqslant C_1,$$

which indicates that $\frac{\Delta_l}{E_l^{\frac{\theta}{\theta-1}}} \leqslant 2^{\frac{\theta}{(\theta-1)^2}} C_1 = 2^{\theta^2\left(\frac{\theta^2-\theta+1}{(\theta-1)^2}+|\log c_1|\right)} := C.$ As a result, we hold a solution with error:

$$F(w_l) - F(w^*) \leqslant C E_l^{\frac{1}{\theta-1}} = C\left(\frac{c^\theta}{\lambda}\right)^{\frac{1}{\theta-1}} \left(\widetilde{R}_{2k,n}\left(\frac{1}{\sqrt{n_l}} + \left(\frac{\sqrt{d \log n_l}}{\epsilon n_l}\right)^{\frac{k-1}{k}}\right) + \frac{\sqrt{\log(1/\beta)}}{2^{p+1}\sqrt{n_l}}\right)^{\frac{\theta}{\theta-1}}$$

$\square$

$\square$

**Proof of Theorem 4.** We first define the set of distributions $\{Q_v\}_{v \in \mathcal{V}}$. Specifically, by the standard Gilbert-Varshamov bound, there exists a set $\mathcal{V} \subset \{\pm\}^d$ such that: (1) $|\mathcal{V}| \geq 2^{\frac{d}{20}}$, (2) for all $v, v' \in \mathcal{V}$, $d_{ham}(v, v') \geq \frac{d}{8}$ (1). For each $v \in \mathcal{V}$, we define $Q_v$ as

$$X_v = \begin{cases} 0, & \text{with probability } 1 - p \\ p^{-\frac{1}{k}} \frac{\tilde{r}_k}{2\sqrt{d}} v, & \text{with probability } p \end{cases} \tag{14}$$

We can see that for each $X_v \sim Q_v$, we always have $\|\mu_v = \mathbb{E}[X_v]\|_2 = p^{\frac{k-1}{k}} \frac{\tilde{r}_k}{2} = \mu.$

We then consider the loss function $f(w, x) = -\langle w, x \rangle + \frac{1}{\theta} \|w\|_2^\theta$, i.e., $F_P(w) = -\langle w, \mathbb{E}_P[x]\rangle + \frac{1}{\theta}\|w\|_2^\theta$ for distribution $P$. By (30) we know it satisfies $(\theta, 1)$-TNC when $\theta \geq 2$. Moreover, for each $Q_v$ we have

$$\mathbb{E}[\sup_{w \in \mathcal{W}} \|\nabla f(w, x)\|_2^k] = \mathbb{E}[\sup_{w \in \mathcal{W}} \|\|w\|_2^{\theta-2} w - x\|_2^k] \leqslant \mathbb{E}[\|2x\|_2^k] = \tilde{r}_k^k = \tilde{r}^{(k)}, \tag{15}$$

where the first inequality is due to the radius of $\mathcal{W}$ is $\left(\frac{p^{-\frac{1}{k}} \tilde{r}_k}{2}\right)^{\frac{1}{\theta-1}}$. Thus we can see $F_P(w)$ satisfies Assumption 1. For convenience we denote $F_{Q_v}(w) = F_v(w)$.

By the form the $F_v(w)$ we can also see that

$$\nabla F_v(w^*) = 0 \equiv \|w^*\|_2^{\theta-2} w^* = \mu_v. \tag{16}$$

Thus the optimal solution $w_v^* = \frac{\mu_v}{\mu^{\frac{\theta-2}{\theta-1}}} \in \mathcal{W}$ by our assumption on $n$ and thus $p \leqslant 1$. In total we have

$$\mathcal{M}(\mathcal{W}, \mathcal{P}, \mathcal{F}_k^\theta(\mathcal{P}, \tilde{r}_k), \rho) \geq \inf_{\mathcal{A} \in \mathcal{Q}(\rho)} \frac{1}{|\mathcal{V}|} \sum_{v \in \mathcal{V}} \mathbb{E}_{\mathcal{A}, D \in Q_v^n}[F_v(\mathcal{A}(D)) - \min_{w \in \mathcal{W}} F_v(w)], \tag{17}$$

$$\geq \inf_{\mathcal{A} \in \mathcal{Q}(\rho)} \frac{1}{|\mathcal{V}|} \sum_{v \in \mathcal{V}} \mathbb{E}_{\mathcal{A}, D \in Q_v^n} \|\mathcal{A}(D) - w_v^*\|_2^\theta = \inf_{\mathcal{A} \in \mathcal{Q}(\rho)} \frac{1}{|\mathcal{V}|} \sum_{v \in \mathcal{V}} \mathbb{E}_{\mathcal{A}, D \in Q_v^n} \|\mathcal{A}(D) - \frac{\mu_v}{\mu^{\frac{\theta-2}{\theta-1}}}\|_2^\theta. \tag{18}$$

Next, we recall the following private Fano's lemma:

**Lemma 6.** *[Theorem 1.4 in (20)] Let $\mathcal{P}$ be a class of distributions over a data universe $\mathcal{X}$. For each distribution $p \in \mathcal{T}$, there is a deterministic function $\theta(p) \in \mathcal{T}$, where $\mathcal{T}$ is the parameter space. Let $\rho : \mathcal{T} \times \mathcal{T} :\mapsto \mathbb{R}_+$ be a semi-metric function on the space $\mathcal{T}$ and $\Phi : \mathbb{R}_+ \mapsto \mathbb{R}_+$ be a non-decreasing function with $\Phi(0) = 0$. We further assume that $X = \{X_i\}_{i=1}^n$ are $n$ i.i.d observations drawn according to some distribution $p \in \mathcal{P}$, and $Q : \mathcal{X}^n \mapsto \Theta$ be some algorithm whose output $Q(X)$ is an estimator. Consider a set of distributions $\mathcal{V} = \{p_1, p_2, \cdots, p_M\} \subseteq \mathcal{P}$ such that for all $i \neq j$,*

- $\Phi(\rho(\theta(p_i), \theta(p_j))) \geq \alpha,$

- $D_{KL}(p_i, p_j) \leqslant \beta$, where $D_{KL}$ is the KL-divergence,

- $D_{TV}(p_i, p_j) \leqslant \gamma$,

*then we have for any $\rho$-zCDP mechanism $Q$.*

$$\frac{1}{M} \sum_{i \in [M]} \mathbb{E}_{X \sim p_i^n, Q}[\Phi(\rho(Q(X), \theta(p_i)))] \geq \frac{\alpha}{2} \max\{1 - \frac{n\beta + \log 2}{\log M}, 1 - \frac{\rho(n^2 \gamma^2 + n\gamma(1-\gamma)) + \log 2}{\log M}\}.$$

Now we will leverage the above lemma to lower bound equation 18. We can see in our set of probabilities $\{Q_v\}_{v \in \mathcal{V}}$, for any $v, v' \in \mathcal{V}$ we have $D_{TV}(Q_v, Q_{v'}) \leqslant p$. And

$$\|\frac{\mu_v}{\mu^{\frac{\theta-2}{\theta-1}}} - \frac{\mu_{v'}}{\mu^{\frac{\theta-2}{\theta-1}}}\|_2^\theta = \frac{1}{\mu^{\frac{\theta(\theta-2)}{\theta-1}}} \|p^{\frac{k-1}{k}} \frac{\tilde{r}_k}{2\sqrt{d}}(v - v')\|_2^\theta \geq C \frac{p^{\frac{\theta(k-1)}{k}}}{\mu^{\frac{\theta(\theta-2)}{\theta-1}}} \tilde{r}_k^\theta = \Omega(\tilde{r}_k^{\frac{\theta}{\theta-1}} p^{\frac{k-1}{k} \frac{\theta}{\theta-1}}). \quad (19)$$

Taking $p = \frac{\sqrt{d}}{n\sqrt{\rho}}$ and by Lemma 6 we have

$$\inf_{\mathcal{A} \in \mathcal{Q}(\rho)} \frac{1}{|\mathcal{V}|} \sum_{v \in \mathcal{V}} \mathbb{E}_{\mathcal{A}, D \in Q_v^n} \|\mathcal{A}(D) - \frac{\mu_v}{\mu^{\frac{\theta-2}{\theta-1}}}\|_2^\theta \geq \Omega\left((\tilde{r}_k(\frac{\sqrt{d}}{n\sqrt{\rho}})^{\frac{k-1}{k}})^{\frac{\theta}{\theta-1}}\right). \quad (20)$$

$\square$

**Proof of Theorem 5.** The lower bound for non-private case follows the proof in (2). Here we extend to the heavy-tailed case. For the index set $\mathcal{V}$ we consider the same one as in the proof of Theorem 4. For each $v \in \mathcal{V}$ we define $X \sim P_v$ as

$$\text{for } j \in [d], X_j = \begin{cases} v_j e_j \frac{\tilde{r}_k}{2\sqrt{d}}, & \text{with probability } \frac{1+\delta}{2}, \\ -v_j e_j \frac{\tilde{r}_k}{2\sqrt{d}}, & \text{with probability } \frac{1-\delta}{2}. \end{cases} \quad (21)$$

We can see that for each $X_v \sim Q_v$, we always have $\|\mu_v = \mathbb{E}[X_v]\|_2 = \delta \frac{\tilde{r}_k}{2} = \mu$.

We then consider the loss function $f(w, x) = -\langle w, x \rangle + \frac{1}{\theta} \|w\|_2^\theta$, i.e., $F_P(w) = -\langle w, \mathbb{E}_P[x] \rangle + \frac{1}{\theta} \|w\|_2^\theta$ for distribution $P$. By (30) we know it satisfies $(\theta, 1)$-TNC when $\theta \geq 2$. Moreover, for each $Q_v$ we have

$$\mathbb{E}[\sup_{w \in \mathcal{W}} \|\nabla f(w, x)\|_2^k] = \mathbb{E}[\sup_{w \in \mathcal{W}} \| \|w\|_2^{\theta-2} w - x\|_2^k] \leqslant \mathbb{E}[\|2x\|_2^k] = \tilde{r}_k^k = \tilde{r}^{(k)}, \quad (22)$$

where the first inequality is due to the radius of $\mathcal{W}$ is $(\frac{\tilde{r}_k}{2})^{\frac{1}{\theta-1}}$. Thus we can see $F_P(w)$ satisfies Assumption 1. For convenience we denote $F_{Q_v}(w) = F_v(w)$.

By the form the $F_v(w)$ we can also see that

$$\nabla F_v(w^*) = 0 \equiv \|w^*\|_2^{\theta-2} w^* = \mu_v. \quad (23)$$

Thus the optimal solution $w_v^* = \frac{\mu_v}{\mu^{\frac{\theta-2}{\theta-1}}} \in \mathcal{W}$ by our assumption on $n$ and thus $p \leqslant 1$. In total we have

$$\mathcal{M}(\mathcal{W}, \mathcal{P}, \mathcal{F}_k^\theta(\mathcal{P}, \tilde{r}_k), \rho) \geq \inf_{\mathcal{A} \in \mathcal{Q}(\rho)} \frac{1}{|\mathcal{V}|} \sum_{v \in \mathcal{V}} \mathbb{E}_{\mathcal{A}, D \in Q_v^n}[F_v(\mathcal{A}(D)) - \min_{w \in \mathcal{W}} F_v(w)], \quad (24)$$

$$\geq \inf_{\mathcal{A} \in \mathcal{Q}(\rho)} \frac{1}{|\mathcal{V}|} \sum_{v \in \mathcal{V}} \mathbb{E}_{\mathcal{A}, D \in Q_v^n} \|\mathcal{A}(D) - w_v^*\|_2^\theta = \inf_{\mathcal{A} \in \mathcal{Q}(\rho)} \frac{1}{|\mathcal{V}|} \sum_{v \in \mathcal{V}} \mathbb{E}_{\mathcal{A}, D \in Q_v^n} \|\mathcal{A}(D) - \frac{\mu_v}{\mu^{\frac{\theta-2}{\theta-1}}}\|_2^\theta. \quad (25)$$

We can see in our set of probabilities $\{Q_v\}_{v \in \mathcal{V}}$, for any $v, v' \in \mathcal{V}$ we have $D_{KL}(Q_v, Q_{v'}) \leqslant \delta^2$. And

$$\|\frac{\mu_v}{\mu^{\frac{\theta-2}{\theta-1}}} - \frac{\mu_{v'}}{\mu^{\frac{\theta-2}{\theta-1}}}\|_2^\theta = \frac{1}{\mu^{\frac{\theta(\theta-2)}{\theta-1}}} \|\frac{\delta \tilde{r}_k}{2\sqrt{d}}(v - v')\|_2^\theta \geq C \frac{\delta^\theta}{\mu^{\frac{\theta(\theta-2)}{\theta-1}}} \tilde{r}_k^\theta = \Omega(\tilde{r}_k^{\frac{\theta}{\theta-1}} \delta^{\frac{\theta}{\theta-1}}). \quad (26)$$

Thus by Fano's lemma or Lemma 6, taking $\delta = \sqrt{\frac{d}{n}}$ we have the result. $\square$

**Proof of Theorem 6.** **Proof of Privacy.** We first recall the following lemma:

**Lemma 7.** *(15) For a domain $\mathcal{D}$, let $\mathcal{R}^{(i)} : f \times \mathcal{D} \to \mathcal{S}^{(i)}$ for $i \in [n]$ be a sequence of algorithms such that $\mathcal{R}^{(i)}(z_{1:i-1}, \cdot)$ is a $(\epsilon_0, \delta_0)$-DP local randomizer for all values of auxiliary inputs $z_{1:i-1} \in \mathcal{S}^{(1)} \times \cdots \times \mathcal{S}^{(i-1)}$. Let $\mathcal{A}_\mathcal{S} : \mathcal{D}^n \to \mathcal{S}^{(1)} \times \cdots \times \mathcal{S}^{(n)}$ be the algorithm that given a dataset $x_{1:n \in \mathcal{D}^n}$, sample a uniformly random permutation $\pi$, then sequentially computes $z_i = \mathcal{R}^{(i)}(z_{1:i-1}, x_{\pi(i)})$ for $i \in [n]$, and the outputs $z_{1:n}$. Then for any $\delta \in [0, 1]$ such that $\epsilon_0 \leqslant \log\left(\frac{n}{16 \log(2/\delta)}\right)$, $\mathcal{A}_\mathcal{S}$ is $(\epsilon, \delta + O(e^\epsilon \delta_0 n))$-DP where $\epsilon = O\left((1 - e^{-\epsilon_0}) \cdot (\frac{\sqrt{e^{\epsilon_0} \log(1/\delta)}}{\sqrt{n}} + \frac{e^{\epsilon_0}}{n})\right)$.*

We know that for each $x \in \mathcal{B}_t$, we have $\mathcal{R}(\Pi_C(\nabla f(w, x))) = \Pi_C(\nabla f(w, x)) + \zeta_x$, with $\zeta_x \sim \mathcal{N}(0, \sigma_1^2)$ and $\sigma_1^2 = \frac{8C^2 \log \frac{1}{\delta_0}}{\epsilon_0^2})$ is an $(\epsilon_0, \delta_0)$-LDP randomizer. As we randomly shuffled the data in the beginning, thus, the algorithm will be $(\hat\epsilon, \hat\delta + O(e^{\hat\epsilon} \delta_0 n))$-DP where $\hat\epsilon = O\left((1 - e^{-\epsilon_0}) \cdot (\frac{\sqrt{e^{\epsilon_0} \log(1/\hat\delta)}}{\sqrt{n}} + \frac{e^{\epsilon_0}}{n})\right)$.

Now, assume that $\epsilon_0 \leqslant \frac{1}{2}$, then $\exists c_1 > 0$, s.t.,

$$\hat\epsilon \leqslant c_1(1 - e^{-\epsilon_0}) \cdot \left(\frac{\sqrt{e^{\epsilon_0} \log(1/\hat\delta)}}{\sqrt{n}} + \frac{e^{\epsilon_0}}{n}\right)$$

$$\leqslant c_1 \cdot \left((e^{\epsilon_0/2} - e^{-\epsilon_0/2}) \cdot \sqrt{\frac{\log(1/\hat\delta)}{n}} + \frac{e^{\epsilon_0} - 1}{n}\right)$$

$$\leqslant c_1 \cdot \left(\left((1 + \epsilon_0) - (1 - \frac{\epsilon_0}{2})\right) \cdot \sqrt{\frac{\log(1/\hat\delta)}{n}} + \frac{(1 + 2\epsilon_0) - 1}{n}\right)$$

$$= c_1 \cdot \epsilon_0 \cdot \left(\frac{3}{2}\sqrt{\frac{\log(1/\hat\delta)}{n}} + \frac{2}{n}\right).$$

Set $\hat\delta = \frac{\delta}{2}$, $\delta_0 = c_2 \cdot \frac{\delta}{e^{\hat\epsilon} n}$ for some constant $c_2 > 0$ and replace $\epsilon_0 = \frac{2\sqrt{2}C\sqrt{\log \frac{1}{\delta_0}}}{\sigma_1}$):

$$\hat\epsilon \leqslant c_1 \cdot \frac{2\sqrt{2}C\sqrt{\log \frac{1}{\delta_0}}}{\sigma_1}) \cdot \left(\frac{3}{2}\sqrt{\frac{\log(1/\hat\delta)}{n}} + \frac{2}{n}\right)$$

$$\leqslant O\left(\frac{C \cdot \sqrt{\log(1/\delta) \log(e^{\hat\epsilon} n/\delta)}}{\sigma_1 \sqrt{n}}\right).$$

For any $\epsilon \leqslant 1$, if we set $\sigma = O\left(\frac{C \cdot \sqrt{\log(1/\delta) \log(e^{\hat\epsilon} n/\delta)}}{\epsilon \sqrt{n}}\right)$, then we have $\hat\epsilon \leqslant \epsilon$. Furthermore, we need $\epsilon_0 = \frac{2\sqrt{2}C\sqrt{\log \frac{1}{\delta_0}}}{\sigma_1}) \leqslant \frac{1}{2}$, which would be ensured if we set $\epsilon = O\left(\sqrt{\frac{\log(n/\delta)}{n}}\right)$. This implies that for $\sigma_1 = O\left(\frac{C \cdot \sqrt{\log(1/\delta) \log(e^{\hat\epsilon} n/\delta)}}{\epsilon \sqrt{n}}\right)$, algorithm 6 satisfies $(\epsilon, \delta)$-DP as long as $\epsilon = O\left(\sqrt{\frac{\log(n/\delta)}{n}}\right)$ if releasing $\mathcal{R}(\Pi_C(\nabla f(w, x)))$ for all $x$. Thus in step 6 we can see $\widetilde{\nabla} F_t\left(w_t^{md}\right) = \frac{T}{n} \sum_{x \in \mathcal{B}_t}(\mathcal{R}(\Pi_C(\nabla f(w_t^{md}, x)))$ is $(\epsilon, \delta)$-DP for each $t$. And since $\{B_t\}$ are disjoint, Algorithm 6 is $(\epsilon, \delta)$-DP.

**Lemma 8.** *(4) Let $\{z_i\}_{i=1}^s \sim \mathcal{D}^s$ be $\mathbb{R}^d$-valued random vectors with $\mathbb{E}z_i = \nu$ and $\mathbb{E}\|z_i\|^k \leqslant r^{(k)}$ for some $k \geqslant 2$. Denote the noiseless average of clipped samples by $\widehat{\nu} := \frac{1}{s}\sum\limits_{i=1}^s \prod_C(z_i)$ and $\widetilde{\nu} := \widehat{\nu} + N$. Then, $\|\mathbb{E}\widetilde{\nu} - \nu\| = \|\mathbb{E}\widehat{\nu} - \nu\| \leqslant \mathbb{E}\|\widehat{\nu} - \nu\| \leqslant \frac{r^{(k)}}{(k-1)C^{k-1}}$, and $\mathbb{E}\|\widetilde{\nu} - \mathbb{E}\widetilde{\nu}\|^2 = \mathbb{E}\|\widetilde{\nu} - \mathbb{E}\widehat{\nu}\|^2 \leqslant d\sigma^2 + \frac{r^{(2)}}{s}$.*

claim: we can improve the noise to $\Sigma^2 := \sup_{t\in[T]} \mathbb{E}[\|N_t\|^2] \leqslant d\sigma^2 + \frac{r^2 T}{n} \approx \frac{dC^2 T}{\epsilon^2 n^2} + \frac{r^2 T}{n}$.

Excess risk: Consider round $t \in [T]$ of Algorithm 6, where Algorithm 1 is run on input data $\{\nabla f(w_t, x_i^t)\}_{i=1}^{n/T}$. Denote the bias of Algorithm 1 by $b_t := \mathbb{E}\widetilde{\nabla} F_t(w_t) - \nabla F(w_t)$, where $\widetilde{\nabla} F_t(w_t) = \widetilde{\nu}$ in the notation of Algorithm 1. Also let $\hat{\nabla} F_t(w_t) := \hat{\mu}$ (in the notation of Lemma 8) and denote the noise by $N_t = \widetilde{\nabla} F_t(w_t) - \nabla F(w_t) - b_t = \widetilde{\nabla} F_t(w_t) - \mathbb{E}\widetilde{\nabla} F_t(w_t)$. Then we have $B := \sup_{t\in[T]} \|b_t\| \leqslant \frac{r^{(k)}}{(k-1)C^{k-1}}$ and $\Sigma^2 := \sup_{t\in[T]} \mathbb{E}\left[\|N_t\|^2\right] \leqslant d\sigma^2 + \frac{r^2 T}{n} \leqslant O\left(\frac{dC^2 T}{\epsilon^2 n^2} + \frac{r^2 T}{n}\right)$, by Lemma 5. Plugging these estimates for $B$ and $\Sigma^2$ into Proposition 40 of (29) and setting $C = r\left(\frac{\epsilon n}{\sqrt{d\log(1/\delta)}}\right)^{1/k}$, we get

$$\mathbb{E}F(w_T^{ag}) - F^* \leqslant O\left(\frac{\beta D^2}{T^2} + \frac{D(\Sigma + B)}{\sqrt{T}} + BD\right)$$

$$\leqslant O\left(\frac{\beta D^2}{T^2} + \frac{CD\sqrt{d\log(1/\delta)}}{\epsilon n} + \frac{rD}{\sqrt{n}} + \frac{r^{(k)}D}{C^{k-1}}\right)$$

$$\leqslant O\left(\frac{\beta D^2}{T^2} + rD\left[\frac{1}{\sqrt{n}} + \left(\frac{\sqrt{d\log(1/\delta)}}{\epsilon n}\right)^{(k-1)/k}\right]\right).$$

Now, our choice of $T$

$$T = \min\{\sqrt{\frac{\beta D}{r}} \cdot \left(\frac{\epsilon n}{\sqrt{d\log(1/\delta)}}\right)^{\frac{k-1}{2k}}, \sqrt{\frac{\beta D}{r}} \cdot n^{1/4}\},$$

implies that $\frac{\beta D^2}{T^2} \leqslant rD\left[\frac{1}{\sqrt{n}} + \left(\frac{\sqrt{d\log(1/\delta)}}{\epsilon n}\right)^{(k-1)/k}\right]$ and we get the result upon plugging in $T$. $\qquad\square$

**Proof of Theorem 7.** Similar to the proof of Theorem 3. $\qquad\square$

