# OpenReview forum: "Private Stochastic Convex Optimization with Tysbakov Noise Condition and Large Lipschitz Constant"
_ICLR.cc/2025/Conference — ICLR 2025 Conference Withdrawn Submission_

### Official Review · Reviewer_qdPg · 2024-11-02

**Soundness:** 3
**Presentation:** 2
**Contribution:** 3
**Rating:** 5
**Confidence:** 3

**Summary:**

This work studies DP-SCO under Tysbakov Noise Condition (TNC, parameterized by $\theta>1$) where the population loss function's gradient is only assumed to have bounded $k>2$-th moment i.t.o. $\ell_2$-norm, which relaxes the common assumption that gradient is uniformly bounded by a Lipschitz constant.
The authors propose several localized noisy clipped gradient methods for different cases with varying $\theta$s.
They show that when applied to convex and smooth functions, these algorithms can achieve faster high-probability convergence rates than that a convex function can achieve.
Moreover, a lower bound under zCDP is derived to confirm the near optimality of the proposed algorithms.
This work is generally interesting and expands our understanding of DP-SCO under TNC.
However, I believe more effort should be put into polishing the manuscript before it is ready for publication.

**Strengths:**

1. the authors considered a setting of TNC that expands our understanding of DP-SCO. The proposed algorithms and theoretical results are promising.
2. a matching minimax lower bound is derived by considering a hard loss function. This function might be of independent interest and may provide insights into the privacy-accuracy tradeoff under TNC settings.

**Weaknesses:**

1. The manuscript seems unfinished. The authors are highly encouraged to conclude the paper with a Conclusion section that summarizes your contributions.
2. There are many typos. Some theorems in the Appendix are not self-contained. This makes readers hard to follow, and they might not be able to fully appreciate the contributions. I will provide more details in Questions.

**Questions:**

1. Some undefined notations & typos:

	(1) Lemma 2 line 244, what is $r^{(k)}$? In the same line, $\tilde{e}_1^k$ should be $\tilde{e}_1^{(k)}$.

 	(2) algorithm 5 line 388, return $w_k$? should be $w_l$

 	(3) line 393, there are two 'algorithm 5'

 	(4) algorithm 7 line 475, what is AC-SA? this algorithm is not mentioned anywhere else.

 	(5) please be as precise as possible with the usage of per-sample loss $f$ and population loss $F$, see lines 807, 863, 873.

2. Theorem 8, 9, 10 are not self-contained. Please properly define all variables used in Theorems.

	(1) In theorem 8, What are $w_T$ and $\hat{w}$? I presume this theorem is associated with algorithm 2, so $w_T$ is the output of algorithm 2. But $\hat{w}$ is still not well-defined. The definition of $b_t$ should appear earlier, as you need it for the eq in line 746.

	(2) In theorem 9, What are $w_l$ and $\hat{w}_l$? In algorithm 3, $w_l$ is used for the output, but in algorithm 4, this output is assigned to $\hat{w}_l$. In this sense, $w_l$ and $\hat{w}_l$ are the same. Please be clear on the meanings of notations used in Theorems. Additionally, please be accurate with the usage of per-sample loss $f$ and population loss $F$.

	(3) similar issues in theorem 10. Please define $w_l$ and $w^*$. Or at least associate it with a specific algorithm

3. Theorem 8 line 805, do you have the assumption $\eta\leq \frac{2}{\lambda}$? is this an implicit result of a properly chosen $p$? I am curious because in algorithm 3 step 6 when you invoke algorithm 2, the strong convexity $\lambda_i$ is parameterized by $p$, it is not immediate to see whether this assumption holds.

4. Equation in line 863 seems incorrect? Even if you replace $f$ with $F$ on the rhs, it seems the $w^*$ term is still missing on the rhs. Do I miss something?

5. A clarification question: I notice the lower bounds are proved for $(\theta, 1)$-TNC while the upper bounds are for $(\theta, \lambda)$-TNC. So there still exists a gap in terms of parameter $\lambda$? To be specific, it seems upper bounds and lower bounds differ in a term of $(\frac{1}{\lambda})^{\frac{1}{\theta-1}}$, if $\lambda\neq1$.

---

### Official Review · Reviewer_YiKD · 2024-11-03

**Soundness:** 3
**Presentation:** 3
**Contribution:** 3
**Rating:** 5
**Confidence:** 3

**Summary:**

The paper explores DP-SCO with large Lipschitz constants under TNC conditions. Key findings include achieving theoretical bounds that are independent of the Lipschitz constant. The paper also provides lower bounds in some settings.This research contributes significantly to the domain of privacy-preserving machine learning, particularly in managing complex data distributions with rigorous privacy requirements.

**Strengths:**

The paper is well-written, with algorithms clearly delineated and accompanied by thorough commentary.

An upper bound that is independent of the Lipschitz constant and a lower bound were derived, demonstrating the algorithm's effectiveness and providing a valuable theoretical tool for further research.

A comparison with previous results on DP-SCO under varying assumptions is provided, positioning this work effectively within the related literature.

**Weaknesses:**

The TNC combined with Assumption 1 (bounded moments and diameter), while justified by related works, may limit the broader interest from the ICLR audience, potentially making it more suitable for a journal. Further explanation of related applications could help address this issue.

The novelty of the algorithm should be more distinctly explained to clarify how it differs from standard "sample, clip, SGD" methodologies.

The lower and upper bounds are not presented within the same framework, understandably due to the absence of amplification results under CDP.

**Questions:**

Could you provide some experimental results regarding the efficiency (speed and actual accuracy) of the proposed method?

The lower and upper bounds are not presented within the same framework. Could you use some conversion between DP, GDP, and CDP to allow for a comparative analysis to some extent?

---

### Official Review · Reviewer_FRW4 · 2024-11-04

**Soundness:** 3
**Presentation:** 2
**Contribution:** 2
**Rating:** 6
**Confidence:** 3

**Summary:**

This paper studies the problem of Stochastic Convex Optimization under the lens of Differential Privacy. Previous papers usually assume the loss is Lipschitz so the sensitivity of the gradient is bounded, thus making it easier to apply Gaussian Mechanism to guarantee privacy. This paper relaxes this Lipschitz assumption and derives a high probability bound for the population risk.

**Strengths:**

- Designing differentially private algorithms without requiring Liptchitz loss is a pretty interesting problem. Though the gradient norm in practice tends to be bounded with high probability, it is usually not uniformly bounded. Thus, it is necessary to develop theories that can cover this realistic scenario.

- The algorithm has good theoretical guarantees. The algorithm basically recovers the optimal asymptotic rate of the previous works. The author also proves a lower bound to show that their guarantee is optimal up to some constants.

- The analysis is sound and correct

- The Tysbakov Noise condition is novel and hasn't been extensively studied in the literature.

**Weaknesses:**

- I think the paper would greatly benefit from some high-level discussion of the algorithm. The author doesn't have to go through every steps of the algorithm but they should highlight some key ideas of the algorithm (they did highlight a few like shuffling, and separating algorithm into phases but it is not clear how those technical ideas contribute to the final results).

- The author claims that their locality algorithm as being novel but as far as I know, this is a pretty common method for Stochastic optimization for strongly convex losses (in their cases, losses that satisfy the Tysbakov Noise Condition). See [1]. Obviously, the actual parameters used and the partitions are different so I would recommend the authors focus on that.

- The paper needs a bit of discussion on the TNC. Seems like it's some generalization of strong convexity but the paper does not provide strong arguments on why we should care about this settings. Some more discussion on this would be greatly appreciated.

- I'm not too convinced that the new constant (that basically replaces the Lipchitz constant in the final bound) is that much smaller than the constant from doing clipped SGD.

- To improve the bound in section 6, the authors use privacy amplification by sampling which results in $\epsilon$ being tiny just for this algorithm to work. This is not exactly a huge surprise since privacy amplification by sampling is more suited for the settings where we take multiple passes over the dataset.

[1] Feldman, Vitaly, Tomer Koren, and Kunal Talwar. "Private stochastic convex optimization: optimal rates in linear time." Proceedings of the 52nd Annual ACM SIGACT Symposium on Theory of Computing. 2020.

**Questions:**

- Can the author compare the results of this paper to the results of [1]. [1] also does not require Lipschiz loss. I think the main reason why the proposed algorithm is able to handle non-Lipschitz loss is basically through clipping.

- Why do we need to switch to $\rho-$zCDP for lower bounds?

- What's the role of the constant $p$?

[1] Chen, Xiangyi, Steven Z. Wu, and Mingyi Hong. "Understanding gradient clipping in private sgd: A geometric perspective." Advances in Neural Information Processing Systems 33 (2020): 13773-13782.

---

> ### Author Response · Authors · 2024-11-20
>
> We believe that the reviewer misunderstood some of our contributions. And we have already make thorough explanation in our paper. Below we will address your concerns.
>
> >**W1**
>
> **Response** Localization involves iteratively "zooming in" on a solution. In the early phases, when we are far from the optimum $w^*$, we use more samples (larger $ n_i $) and a larger learning rate $\eta_i$ to make quick progress. As $i$ increases and $ w_i $ gets closer to $w^*$, fewer samples and a slower learning rate suffice. Since the step size $\eta_i$ shrinks geometrically faster than $n_i$, the effective variance of the privacy noise ($\eta_i^2\sigma_i^2$) decreases with $ i $. This prevents $w_{i+1}$ from deviating too far from $w_i$ and hence from $w^*$. We further enforce this localization behavior by increasing the regularization parameter $\lambda_i$ and shrinking $ D_i $ over time. Additionally, we used the minimizer from the previous phase as the initialization for the subsequent phase.  We choose $D_i$ to be as small as possible while ensuring that $argmin_{w\in\mathcal{W}} \widehat{F}_i(w)\in\mathcal{W}_i$. This constraint ensures that Algorithm 2 can find $ w_i $ with minimal excess risk and also leads to the final estimation for $w_l$. Details in Line 740, Theorem 8 show how the tiny diameters and stepsizes contribute to the estimation. Especially, Line 794 shows how iteration helps.
>
> In Algorithm 4, the updates are divided into m stages, and we perform LNC-GM at each stage. Each use of Alg3 is initialized from the endpoint obtained from the previous stage， inheriting some property from last stage. Line 950, Lemma 5 shows what we do exactly in each step.
>
> In Algorithm 6, we randomly permute each batch before executing the main algorithm, which helps reduce variance. This technique, combined with the methods mentioned above, allows us to improve the upper bound in Theorem 6. In line 6, our clipping method eliminates the need for the Lipschitz property of loss functions.
>
> Similar iterated idea is introduced into Algorithm 5 and 7. For Algorithm 5, we partition the dataset aggressively while keeping the convex set to be projected invariant, which is necessary, for instance, in Line 1079 and from Line 1088 to 1108, proof of theorem 3.
>
> >**W2**
>
> **Response** **We cannot agree with the comment; our algorithm is totally different from the localization algorithm (Algorithm 2) of [1]**:
>
> (1) The locality algorithm in [1] is based on the stability of projected SGD, meaning that it depends on the Lipschitz condition, and thus, all of their results depend on the  Lipschitz constant. However, as we mentioned, our paper aims to eliminate this constant and we aim to use the moment instead. Thus, our algorithm (Algorithm 3) is more complicated.
>
> (2) The locality algorithm in [1]  is for general convex loss while we consider the TNC loss. However, directly using the stability of SGD cannot leverage such property. Thus, we need to design a new locality algorithm (our Algorithm 3).
>
> (3) The details and the main idea of our Algorithm 3 are also different from the locality algorithm in [1]. Specifically, for each phase, we need to consider a regularized function (step 5) and perform a clipped version (the clipping threshold is also important) of the gradient descent method over a shrunk domain set $\mathcal{W}_i$. These are crucial for our analysis. However, in the locality algorithm of [1], during each phase, it just simply performs the projected gradient descent over the original domain set.
>
> Thus, these two algorithms are incomparable.

---

> ### Author Response · Authors · 2024-11-20
> **We will continue to add explanation here.**
>
> >**W3**
>
> **Response**  Thanks for the suggestion. In fact, we are motivated by the fact that while most of the current paper focuses on the loss as either convex or strongly convex, many losses lie between these two categories, i.e., they are not strongly convex but their statistical rate is better than convex ones. So, we want to study the theoretical behaviors of these loss functions.
>
> It has been shown that many machine learning problems beyond the example we provided are not strongly convex but satisfying TNC, which also motivates this work [1,33]. For example:
>
> (1) Piecewise Linear Problems (PLP): $\min_{W}\mathbb{E} [f(w,x)]$. When the objective function is piecewise linear (such as hinge loss) and W is a bounded polyhedron. Then it satisfies $(2, O(1))$-TNC.
>
> (2) Risk Minimization Problems over an $\ell_2$ ball. $\min_{W}\mathbb{E} [f(w,x)]$. When the W is an $\ell_2$-norm ball and $\min_{W}\mathbb{E} [f(w,x)]< \min_{\mathbb{R}^d}\mathbb{E} [f(w,x)]$. Then it satisfies $(2, O(1))$-TNC.
>
> (3) $\ell_1$ Regularized Risk Minimization Problems. For $\ell_1$ regularized problem $\min_{\|w\|_1\leq B}\mathbb{E} [f(w,x)]+\lambda \|w\|_1$, it satisfies $(2, O(1))$-TNC.
>
> Thus, designing appropriate algorithms will give these problems better generalization ability.
>
> Moreover previous studies on TNC always need to assume the data are regular, such as normalized, and they cannot handle cases where these data are heavy-tailed. Thus, it is essential to design algorithms to handle this heavy-tailed and sensitive data.
>
> [1] Liu, Mingrui, et al. "Fast rates of erm and stochastic approximation: Adaptive to error bound conditions." Advances in Neural Information Processing Systems 31 (2018).
>
> In the final version, we will add the above motivations to the introduction section.
>
>
> >**W4**
>
> **Response**  **We disagree with the comment.**
> First, to the best of our knowledge, we still do not know the **excess population risk given by clipped SGD with general clipping threshold** if the convex loss function is not Lipschitz. We would appreciate it if the reviewer could provide some references, and we would be happy to compare them. Most existing work on the excess population risk of clipped SGD needs to assume the loss function is Lipschitz, and they assume the clipping threshold is the Lipschizt constant, indicating the utility will depend on the Lipschitz constant. From this perspective, our parameter could be much smaller than the Lipschitz constant, and thus, our results are better.
>
> Here we provide an example to show that the moment bound  could be far less than the Lipschitz constant:
> For linear regression on a unit ball W with 1-dimensional data $x_1,x_2\in[-10^{10},10^{10}]$ having Truncated Normal distributions and $Var(x_1)=Var(x_2)\leq 1$, we have $L_f\geq 10^{20}$ while for $k<\infty$, $\tilde{r}_k$ is much smaller than $L_f$, such as $\tilde{r}_2\leq 5,\tilde{r}_4\leq 8,\tilde{r}_8\leq 14$.
>
> >**W5**
>
> **Response** The review misunderstood our algorithm, **note that here we use privacy amplification by shuffling rather than subsampling**. Privacy amplification by sampling cannot achieve our results. Moreover, the comment "which results in $\epsilon$ being tiny just for this algorithm to work" is not correct. The requirement of $\epsilon$ comes from the current theorem of privacy amplification by shuffling.  As we mentioned in the paper, we showed that using shuffling can significantly improve the previous results. Our algorithm can also remove the Lipschitz assumption.
>
> >**Q1**
>
> **Response** Our results are totally different from Chen et al [1]:
>
> (1) The problem studied is different. In our paper we focus on DP-SCO for TNC loss and aim to develop the optimal rates of excess population risk. [1] aims to understand clipped DP-SGD for general (non)-convex loss. It is unknown whether clipped DP-SGD can achieve the optimal rate for our problem. In our paper, we develop several new algorithms for optimality.
>
> (2) The utility form is also different. We mainly use excess population risk for the utility, which is the standard utility in the existing literature on DP-SCO or DP-ERM with convex loss. [1] aims to provide a convergence rate for clipped DP-SGD. However, their utility is based on the gradient norm or the inner product with the gradient, and their upper bounds (their Theorem 1 and 5) are incomparable with our results. Actually, [1] mainly focuses on providing some theoretical understanding rather than the excess population risk for the DP-SCO problem. Obviously, their analysis is also totally different from ours.

---

> > ### Comment · Reviewer_FRW4 · 2024-11-23
> >
> > Thank you for your response! The authors have resolved some of my questions and I'll change my score to 6.

---

> ### Author Response · Authors · 2024-11-20
> **Continuation of explanation.**
>
> >**Q2**
>
> **Response** We have already make clarification in Remark 1 after we give the definition of zCDP.  Here we will explain it again. The main reason here is that in the proof, we need a private version of Fano's lemma (Lemma 6 in our paper). Currently, there are two versions of the private Fano's lemma: one is in the zCDP version and the other one is in the $(\epsilon, \delta)$-DP version [2]. However, compared to the zCDP version, the $(\epsilon, \delta)$-DP version is quite loose for our problem. That is the main reason we choose $\rho$-zCDP in the paper. Notably, as we mentioned, our Algorithms 1-5 are based on the stability analysis and Gaussian mechanism, and they are a one-pass algorithm with no privacy amplification. Thus, these algorithms can be converted to the zCDP version directly.
>
> [2] Acharya, Jayadev, Ziteng Sun, and Huanyu Zhang. "Differentially private assouad, fano, and le cam." Algorithmic Learning Theory. PMLR, 2021.
>
> >**Q3**
>
> **Response**
> >Note that $p$ is very important in our algorithm and analysis. We cannot arbitrarily set $p$, it should satisfy $L_f\leqslant n^{p/2}\widetilde{R}_{2k,n}(\frac{1}{\sqrt{n}}+(\frac{\sqrt{d\log n}}{\epsilon n})^{\frac{k-1}{k}})$. Intuitively,
> we explore the potential of using terms like $n^p$ to approximate the magnitude of this constant. The exponent p  is employed to regulate the Lipschitz constant, allowing us to omit it from the final bound. As shown in line 444 of the proof, there is an occurrence of the term $n^p$ in both the numerator and the denominator. By assuming that $L_f$ is governed by $O(n^{p/2})$ and selecting an appropriate $\eta$, we are able to remove the dependence on p in the final bound.

---

### Official Review · Reviewer_WJ5h · 2024-11-05

**Soundness:** 1
**Presentation:** 2
**Contribution:** 1
**Rating:** 3
**Confidence:** 4

**Summary:**

Stochastic optimization has been studied under the privacy constraint by a large body of recent work. Many of the works assume that the loss functions are lipschitz, and some recent works have studied this problem under the weaker assumption of bounded moments of gradient norms. This problem has also been studied under both the convexity assumption as well as a strong convexity assumption.

The current work extends this to losses satisfying a different growth condition known as the Tsybakov noise condition. Intuitively, it says that the function grows at least as fast as a \theta power of the distance from the optimum.  \theta = 1 gives convexity and \theta=2 gives strong convexity. This work considers a range of theta and proposes algorithms which lead to rates that interpolate between these two, and extrapolate beyond theta=2 as well.

I think the problem is interesting. This work shows the first results under this noise condition, and shows that the results are tight.

Unfortunately, I have some concerns about the correctness of the results. Theorem 1 in the paper relies on stability of clipped SGD, which will need a new argument. The authors seem to build on a result about stability of SGD, but clipping of gradients as done in their algorithm may break this stability.

**Strengths:**

- Combines Tsybakov assumption and gradient moment assumptions, and shows new upper and lower bounds for private optimization under these assumptions.

**Weaknesses:**

Weaknesses:
- It is unclear to me why Theorem 1 is true. The proof of privacy in the appendix makes claims about stability. However, this will need a proof: while gradient descent steps are contractive (or strongly contractive in the case of strongly convex losses), the steps taken in Alg 2 are not gradient steps but rather clipped gradient steps. Clipped gradients do not in general correspond to gradients of a convex function, and so it is not clear to my why this argument works.  In fact it seems to me that there is counter-example showing that for strong convexity, strong contractivity does not hold even in one dimension: if f(x)=x^2/2, and you clip gradient norm at 1, then a gradient step at y or y+1 for y> 1 will move to (y-\eta) and (y+1-\eta) respectively. Thus the desired strict contractivity does not seem to hold. Please also see “Characterizing Private Clipped Gradient Descent on convex generalized linear problems” by Song, Thakkar and Thakurta, which shows that convexity can break down.
- Relatedly, If I understand correctly, Thm. 3.9 in HRS has an L^2 in the numerator that seems to be missing in your restatement (Lemma 3). Perhaps there is some normalization difference that I am missing?
- Tsybakov is misspelled in the title and abstract.
- Private Iterative localization based on stability was previously used in [14]. Could you say more on how your use is different from their?
- second line in table seems to be under the same conditions as the first. Is that supposed to be for the strongly convex case? Is there a gap between upper and lower bounds for that case?

Nits:
- Many of the citations of fairly old published work is to the arxiv version. While it may be useful to point to an arxiv version, I would recommend citing the published version.
- Table text is much too small to read.

**Questions:**

See weaknesses above. In particular, I would be curious if you can give a full proof of privacy in Thm. 1.

---

### Note · Authors · 2024-11-27

I have read and agree with the venue's withdrawal policy on behalf of myself and my co-authors.